# Structure and reconstitution of yeast Mpp6-nuclear exosome complexes reveals that Mpp6 stimulates RNA decay and recruits the Mtr4 helicase

Elizabeth V Wasmuth[1], John C Zinder[1,2], Dimitrios Zattas[1], Mom Das[1], Christopher D Lima[1,3]*

[1]Structural Biology Program, Sloan Kettering Institute, Memorial Sloan Kettering Cancer Center, New York, United States; [2]Tri-Institutional Training Program in Chemical Biology, Memorial Sloan Kettering Cancer Center, New York, United States; [3]Howard Hughes Medical Institute, Memorial Sloan Kettering Cancer Center, New York, United States

*For correspondence: limac@mskcc.org

Competing interests: The authors declare that no competing interests exist.

**Abstract** Nuclear RNA exosomes catalyze a range of RNA processing and decay activities that are coordinated in part by cofactors, including Mpp6, Rrp47, and the Mtr4 RNA helicase. Mpp6 interacts with the nine-subunit exosome core, while Rrp47 stabilizes the exoribonuclease Rrp6 and recruits Mtr4, but it is less clear if these cofactors work together. Using biochemistry with *Saccharomyces cerevisiae* proteins, we show that Rrp47 and Mpp6 stimulate exosome-mediated RNA decay, albeit with unique dependencies on elements within the nuclear exosome. Mpp6-exosomes can recruit Mtr4, while Mpp6 and Rrp47 each contribute to Mtr4-dependent RNA decay, with maximal Mtr4-dependent decay observed with both cofactors. The 3.3 Å structure of a twelve-subunit nuclear Mpp6 exosome bound to RNA shows the central region of Mpp6 bound to the exosome core, positioning its Mtr4 recruitment domain next to Rrp6 and the exosome central channel. Genetic analysis reveals interactions that are largely consistent with our model.

## Introduction

The eukaryotic RNA exosome is a conserved and essential multisubunit complex that carries out diverse functions in RNA metabolism, ranging from normal turnover, processing and maturation, to quality control (*Kilchert et al., 2016*; *Zinder and Lima, 2017*). In yeast, its 9-subunit core (Exo9) forms a donut-shaped scaffold that recruits two 3' to 5' exoribonucleases, Dis3 (aka Rrp44) and Rrp6. Subunit composition of RNA exosomes varies according to cellular localization, where Exo9 and Dis3 form the cytoplasmic exosome (Exo10$^{Dis3}$), and Exo9, Dis3, and Rrp6 constitute the nuclear exosome (Exo11$^{Dis3/Rrp6}$). Additionally, human Exo9 and Rrp6 may form a nucleolar exosome (Exo10$^{Rrp6}$) (*Tomecki et al., 2010*). Nuclear RNA exosome activities are associated with an astounding repertoire of RNA substrates, including precursors of ribosomal RNA, snoRNAs, snRNAs, surveillance of tRNA and pre-mRNAs, and decay of non-coding RNAs including CUTs, PROMPTs, and eRNAs (*Allmang et al., 1999, 2000*; *Basu et al., 2011*; *Houseley et al., 2006*; *Pefanis et al., 2015*; *Schneider et al., 2012*). Not surprisingly, mutations in exosome subunits and its misregulation have been reported in several diseases (*Fasken et al., 2017*; *Lohr et al., 2014*; *Robinson et al., 2015*; *Wan et al., 2012*). How the nuclear exosome carries out such diverse processes is thought to be determined in part through its association with regulatory protein cofactors that enhance its ability to interact with appropriate substrates.

Previous studies elucidated structures and biochemical activities of the non-catalytic core and minimal catalytic units: 10-, 11- and 12- subunit exosomes that represent the Exo10$^{Dis3}$ and Exo11-$^{Dis3/Ski7}$ cytoplasmic complexes, the nucleolar Exo10$^{Rrp6}$ complex, and nuclear complexes Exo11$^{Dis3/}$$^{Rrp6}$ and Exo12$^{Dis3/Rrp6/Rrp47}$ (*Liu et al., 2006*; *Wang et al., 2007*; *Kowalinski et al., 2016*; *Makino et al., 2013*, *2015*; *Wasmuth et al., 2014*; *Wasmuth and Lima, 2012*; *Zinder et al., 2016*). Specifically, these studies revealed that Exo9 modulates the activities of Dis3 and Rrp6; that the Rrp6 protein can stimulate Dis3 (*Liu et al., 2006*; *Wasmuth and Lima, 2012*; *Wasmuth et al., 2014*); that the major path for RNAs greater than 30 nucleotides utilize the Exo9 central channel to guide RNA to Rrp6 for distributive trimming or through the entire channel to Dis3 for processive decay (*Bonneau et al., 2009*; *Drazkowska et al., 2013*; *Malet et al., 2010*; *Wasmuth and Lima, 2012*); and the existence of a minor channel-independent 'direct access' route to Dis3 for shorter RNAs (*Liu et al., 2014*; *Makino et al., 2015*; *Zinder et al., 2016*).

While much has been learned about these exosome complexes, an incomplete picture remains with respect to nuclear exosome cofactors and their influence on exosome activities. The nuclear exosome is associated with the protein cofactors Mpp6 and Rrp47, whose individual deletion in vivo results in misprocessing of nuclear exosome substrates, while concomitant loss leads to synthetic lethality (*Butler and Mitchell, 2011*; *Feigenbutz et al., 2013a*; *Milligan et al., 2008*). These genetic observations resemble deletions of the TRAMP complex, a three protein complex that catalyzes pol-yadenylation and subsequent degradation of an array of nuclear RNAs (*LaCava et al., 2005*; *Vanácová et al., 2005*; *Wyers et al., 2005*) through one of two non-templated polyA polymerases Trf4/5, one of two zinc knuckle proteins Air1/2, and the essential RNA helicase Mtr4, which is thought to bridge the nuclear exosome to other RNPs, including ribosome biogenesis factors (*Thoms et al., 2015*). Indeed, Rrp47 binds directly to Rrp6 via the Rrp6 N-terminal domain, an inter-action that stabilizes Rrp6 (*Dedic et al., 2014*; *Feigenbutz et al., 2013a*, *2013b*; *Stead et al., 2007*) and generates a composite interface that recruits Mtr4 (*Schuch et al., 2014*) in vitro and in vivo, though other factors are speculated to exist (*Feigenbutz et al., 2013b*; *Schuch et al., 2014*; *Stuparevic et al., 2013*). In contrast, Mpp6 function is less clear, due in part to its poor sequence conservation. Its synthetic lethality with Rrp6 in yeast suggested Mpp6 may be a Dis3 cofactor (*Milligan et al., 2008*), while in human it was posited as a Rrp6 cofactor (*Schilders et al., 2005*) and has been reported to recruit Mtr4 (*Chen et al., 2001*; *Lim et al., 2017*; *Lubas et al., 2011*; *Schilders et al., 2007*).

To better understand how Mpp6 influences nuclear exosome activities and Mtr4 recruitment, we present the functional characterization of nuclear exosome complexes with Mpp6, Rrp47 and Mtr4 from the budding yeast *Saccharomyces cerevisiae* and the crystal structure of a 12-subunit nuclear exosome complex bound to Mpp6 and RNA to a resolution of 3.3 Å. Using full-length proteins, we observe that Rrp47 and/or Mpp6 can stimulate Rrp6 activity when these factors are associated in exosome complexes, albeit with different dependencies on exosome subunit and domain composi-tion. Finally, biochemical and genetic studies suggest that Mpp6 can bind RNA and stimulate the activities of the nuclear exosome, and that both Mpp6 and Rrp47 contribute to recruitment of the Mtr4 helicase to facilitate activation of the nuclear exosome and degradation of structured RNA.

## Results

### Mpp6 can stimulate Rrp6 activity in an Exo9-dependent manner

Recombinant Mpp6 was expressed, purified and tested for its ability to bind various sequences of single-stranded RNA via fluorescence polarization (*Figure 1—figure supplement 1A*; *Figure 1D*). Similar to previous results (*Milligan et al., 2008*; *Schilders et al., 2005*), yeast Mpp6 bound flexible RNAs most tightly, with a $K_d$ of 170 nM for polyU RNA, and a $K_d$ of 62 nM for AU-rich RNA. Mpp6 also bound polyA RNA, albeit much weaker ($K_d$ of 1.4 µM). Analytical gel filtration was next employed with different components of the exosome core (Exo9) to determine the identity of subu-nits or complexes required for interaction with Mpp6. Consistent with previous results (*Schuch et al., 2014*), Mpp6 interacts with Exo9 (*Figure 1—figure supplement 1B*).

We next assayed Mpp6 effects on degradation of single-stranded polyA RNA, a model substrate for the nuclear exosome, to determine if Mpp6 altered activities of the exosome (*Liu et al., 2006*; *Wasmuth and Lima, 2012*). Mpp6 was mixed with reconstituted exosomes containing Rrp6 that

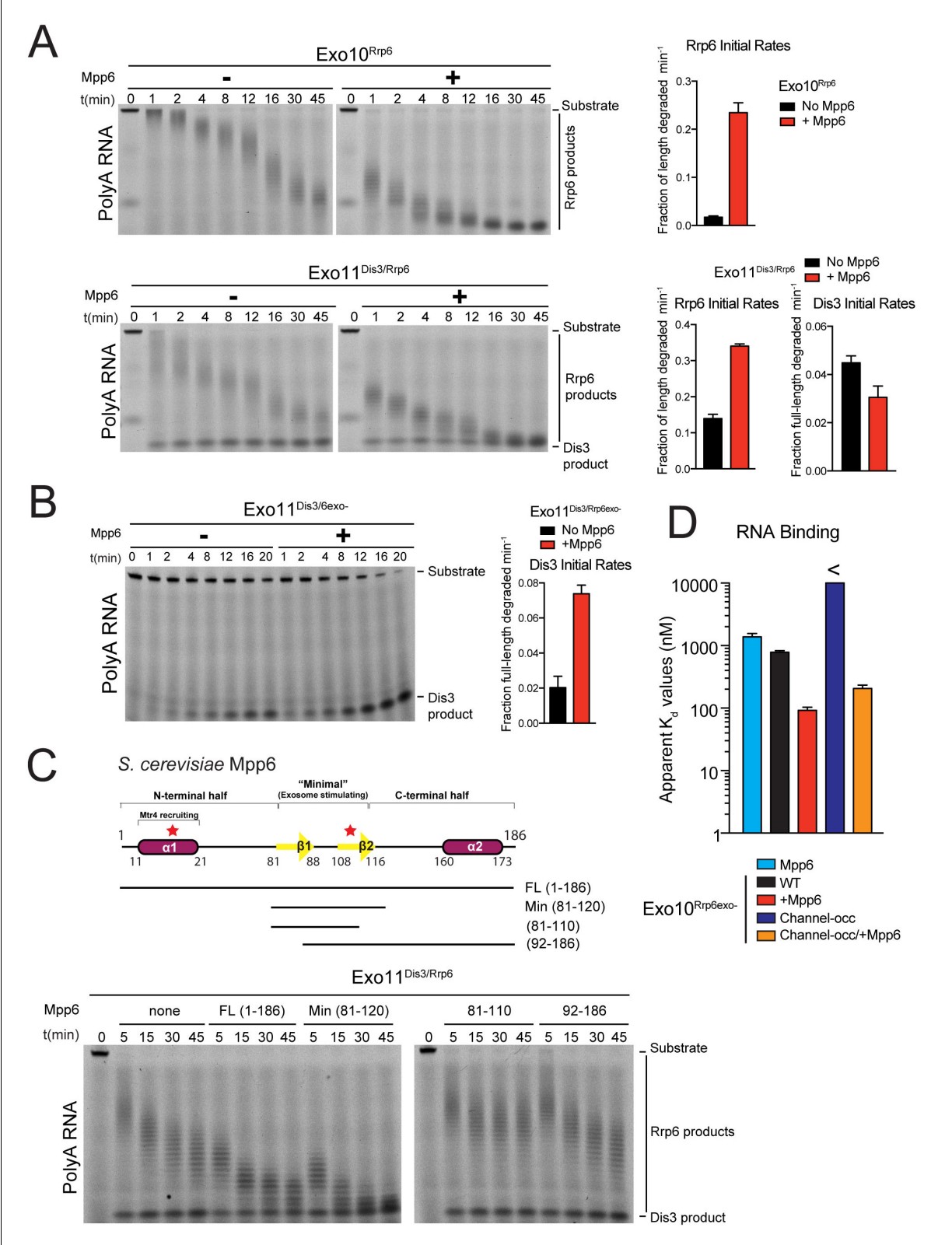

**Figure 1.** Mpp6 stimulates the nuclear RNA exosome and binds RNA. (**A**) Mpp6 stimulates Rrp6 activities when degrading 5' fluorescein-labeled 49 nt polyA in 10- (top) and 11- subunit exosomes (bottom). Relative positions of RNA substrate, Rrp6 products and the Dis3 product are indicated to the right of gels in panels A, B, and C. Quantitation of initial rates shown to the right of representative gels. For Rrp6 initial rates, the median length of Rrp6 products was determined at the earliest time points and used to calculate initial rates using the equation (1 – (median product length/substrate length)/

*Figure 1 continued on next page*

*Figure 1 continued*

min) to yield a fraction of length degraded per minute. For Dis3, initial rates were determined by calculating the fraction of full-length substrate degraded based on the accumulation of 4–5 nt product. For the lower gel, Dis3 and Rrp6 initial rates were determined at the earliest time points where distributive products of Rrp6 are easily distinguished/separated from the 4–5 nt processive products of Dis3. (B) Dis3 exoribonuclease activity can be stimulated on 5′ fluorescein-labeled 49 nt polyA in 11-subunit exosomes if Rrp6 is present but catalytically inert. (C) Top: Predicted domain structure of *S. cerevisiae* Mpp6. Calculated with Jpred (*Cole et al., 2008*). Previously identified conserved regions (*Milligan et al., 2008*) are marked with red stars, with Mtr4 recruiting and exosome stimulating domains as described in this work labeled. Below: a minimal fragment of Mpp6 (residues 81 to 120) is necessary and sufficient to stimulate Rrp6 activity in Exo11$^{Dis3/Rrp6}$. Representative decay assays of Exo11$^{Dis3/Rrp6}$ on 5′ fluorescein-labeled polyA 49 nt RNA with different Mpp6 constructs added in 2-fold molar excess. (D) Mpp6 enhances RNA binding of Rrp6-containing exosomes on polyA RNA, and alleviates binding defects caused by Exo9 channel occlusions. Fluorescence polarization of Mpp6 and Exo10$^{Rrp6exo-}$ with or without PH-like ring occlusions in the lower half of the Exo9 channel, with binding to 5′ fluorescein-labeled polyA 37 nt RNA. Bar graphs and error bars in panels A, B, and D are the result of triplicate experiments with error bars indicating plus or minus one standard deviation.

The following figure supplement is available for figure 1:

**Figure supplement 1.** Mpp6 is a RNA-binding protein cofactor of the nuclear RNA exosome that stimulates exosome exoribonuclease activities.

lacked the PMC2NT domain (128-733) to prevent aggregation, Dis3, or both Rrp6 and Dis3. In the context of Exo10$^{Rrp6}$, Mpp6 stimulated Rrp6-mediated decay 13-fold (*Figure 1A*). Mpp6 stimulated Rrp6 activity by 2.5-fold while reducing Dis3 activity by 1.5-fold when associated with Exo11$^{Dis3/Rrp6}$ (*Figure 1A*). In contrast, Mpp6 had no effect on the activities of Rrp6 or Dis3 in isolation (*Figure 1—figure supplement 1C and D*) and Mpp6 failed to stimulate Dis3 exoribonuclease or endoribonuclease activity on polyA RNA when associated with Exo10$^{Dis3}$ (*Figure 1—figure supplement 1E*). However, Mpp6 could stimulate Dis3 exoribonuclease activity by 3.5-fold in Exo11$^{Dis3/Rrp6exo-}$ when a catalytically inert mutant of Rrp6 (Rrp6$^{exo-}$) was present (*Figure 1B*). Mpp6 had no detectable effect on Dis3 endoribonuclease activity (*Figure 1—figure supplement 1E*). To determine if Mpp6 altered exosome decay activities on another RNA substrate, exoribonuclease assays were conducted using AU-rich RNA (*Liu et al., 2006*; *Wasmuth and Lima, 2012*). Although Mpp6 alone bound AU-rich RNA over 20 times better than polyA RNA (*Figure 1—figure supplement 1A*), Mpp6 stimulated Exo10$^{Rrp6}$ activity by only 1.6-fold using this RNA (*Figure 1—figure supplement 1F*). In contrast, Mpp6 exerted a greater effect on Exo10$^{Dis3}$ for AU-rich RNA, with a 4-fold increase in exoribonuclease activity and 9.5-fold tighter binding for AU-rich RNA (14 nM versus 130 nM) (*Figure 1—figure supplement 1G*). We were unable to assess the influence of Mpp6 on degradation of AU-rich RNA by Exo11$^{Dis3/Rrp6}$ as the RNA was degraded too quickly to measure rates. Taken together, these results suggest that Mpp6 association with Exo9 can stimulate Rrp6 activity, although stimulation of Dis3 can also be observed in the absence of Rrp6 or in the presence of catalytically inactivated Rrp6.

Mpp6 contains two small regions of high sequence conservation as noted previously (*Milligan et al., 2008*), one near its N-terminus and one near the middle of the protein. To define a minimal region of Mpp6 necessary for stimulation of Rrp6, N- and C-terminal truncations of Mpp6 were generated and assayed for their ability to interact with the exosome and to stimulate Exo10$^{Rrp6}$ and Exo11$^{Dis3/Rrp6}$ decay activities using polyA RNA. Activation of Rrp6 in Exo11$^{Dis3/Rrp6}$ only required a central region within Mpp6 that included residues 81–120 (*Figure 1C*), hereafter referred to as 'Mpp6$^{Minimal}$', as other regions were dispensable for activation of Rrp6 in exosomes.

## RNA path to Rrp6 in Exo10$^{Rrp6}$ in the presence of Mpp6

To determine if Rrp6 activity remained dependent on RNA contacts to the Exo9 central channel in the presence of Mpp6, a route of ingress to Rrp6 observed biochemically and structurally in the absence of this nuclear cofactor (*Wasmuth et al., 2014*; *Zinder et al., 2016*; *Wasmuth and Lima, 2012*), exosomes bearing channel mutations within the PH-like ring (lower half; -occ) or the S1/KH ring (upper half) were assayed on polyA RNA with or without Mpp6. Binding measurements could not be reliably determined for Exo10$^{Rrp6exo-/Channel-occ}$ (>10 µM); however, Exo10$^{Rrp6exo-/Mpp6/Channel-occ}$ bound polyA RNA with a $K_d$ of 210 nM, 3.8 fold tighter than Exo10$^{Rrp6exo-/WTchannel+}$ (780 nM), but 2.3 times weaker than Exo10$^{Rrp6exo-/Mpp6/WTchannel+}$ (92 nM) (*Figure 1D*). As Mpp6 on its own binds polyA RNA with a $K_d$ of 1.4 µM (*Figure 1D*; *Figure 1—figure supplement 1A*), it appears that

Mpp6 cooperates with elements within the exosome to bind RNA more tightly in a manner that is partially dependent on the integrity of the Exo9 central channel.

To determine if Mpp6 exosomes utilize a distinct channel-independent route for RNA ingress, or if Mpp6 renders Rrp6 exosomes less dependent on the PH-like ring, we next queried whether Mpp6 could suppress effects of mutations within the S1/KH ring, which have been shown previously to inhibit Rrp6 exoribonuclease activity in exosomes (*Wasmuth et al., 2014*). Although Mpp6 partially rescued defects of these S1/KH ring mutations, the complexes were less active when compared to wild-type Exo10$^{Rrp6}$ (*Figure 1—figure supplement 1H*), suggesting that Mpp6-bound exosomes use a similar RNA path to that observed previously to guide RNA to Rrp6 (*Wasmuth et al., 2014*; *Wasmuth and Lima, 2012*; *Zinder et al., 2016*).

Data presented in *Figure 1* and *Figure 1—figure supplement 1* support a model in which Mpp6 binds the Exo9 core and stimulates the activities of the nuclear exosome, most likely through its ability to bind RNA. While stimulation of Dis3 by Mpp6 was observed in cases where Rrp6 was absent or catalytically inert, the 13-fold stimulation of polyA RNA decay by Rrp6 is among the more striking biochemical effects observed, and is perhaps consistent with the proposed role of Rrp6 in deadenylation of nuclear transcripts in *S. cerevisiae* (*Assenholt et al., 2008*; *Schmid et al., 2012*).

## Mpp6 is anchored to Exo9 via contacts to the S1/KH ring subunit Rrp40

The crystal structure of Exo12$^{Dis3exo-endo-/Rrp6exo-/Mpp6Min}$ was determined to a resolution of 3.3 Å (*Table 1*, *Figure 2A*) using a synthetic RNA with two 3′ ends as reported previously (*Zinder et al., 2016*). Analogous to the previous structure, one RNA 3′ end is anchored in the Dis3 active site with Dis3 adopting a 'direct access' conformation (*Han and van Hoof, 2016*; *Liu et al., 2014*; *Makino et al., 2015*; *Zinder et al., 2016*) while the other 3′ end is anchored in the Rrp6 active site of an adjacent complex (*Zinder et al., 2016*). Although only 4–5 residues of RNA can be visualized in the Rrp6 active site, noncontiguous densities were observed proximal to residues previously identified to contact RNA, including Arg110 of Rrp40, and Rrp6 residues within the EXO and HRDC domains, in particular Tyr244, Tyr430, and Arg461 (*Makino et al., 2015*; *Zinder et al., 2016*).

Electron densities for Mpp6 residues 90 to 118 were evident on the surface of Rrp40 (*Figure 2—figure supplement 1A*). Mpp6$^{Minimal}$ residues 90–97 adopt a distorted parallel beta-strand configuration that complements the second beta-strand of the Rrp40 N-terminal domain (NTD) while residues 98–103 adopt a helical conformation that wedges between the Rrp40 NTD and S1 domain (*Figure 2B*). Mpp6 residues 110–118 adopt an anti-parallel beta-strand conformation that complements the second beta-strand of the KH domain (*Figure 2B*, *Figure 2—figure supplement 1A*). The amino acid register was confirmed by inspection of anomalous data obtained for a low resolution crystal structure of Exo12$^{Dis3endo-exo-/Rrp6exo-/Mpp6Min}$ reconstituted with a selenomethionine substituted Mpp6$^{Minimal}$ that included a single methionine substitution (I94M) (*Figure 2—figure supplement 1A*). Although modeled, densities for amino acids 104–107 were very weak in comparison to other portions of Mpp6. Contacts observed in our structure are also consistent with crosslinking and mass spectrometry data reported previously, including crosslinks between Rrp40 Lys176 and Mpp6 Lys113 and between Rrp40 Lys49 and Mpp6 Lys104 (*Shi et al., 2015*).

Interactions between Rrp40 and Mpp6 are mediated by Rrp40 surfaces with comparable conservation to those involved in contacts to Rrp45 and Rrp46 or RNA (*Malet et al., 2010*; *Wasmuth et al., 2014*; *Zinder et al., 2016*) (*Figure 2C*). Consistent with the structural data, exosomes lacking Rrp40 could not associate with Mpp6, and Mpp6 did not stimulate Rrp6 activity in exosomes lacking Rrp40 (*Figure 2—figure supplement 1B,C*). Although Mpp6 interacts directly with Rrp40 in the context of the exosome complex, interactions could not be detected between Rrp40 and Mpp6 in isolation (*Figure 2—figure supplement 1D*). Furthermore, Mpp6 residues 81 to 90, which are disordered in our structure, are required for exosome Rrp6 stimulation (*Figure 1C*).

Mpp6 residues Leu92, Ile93 and Ile94 reside in the first beta strand and are nestled into a hydrophobic surface of the Rrp40 NTD (*Figure 3B*). The hydrophobic quality of these Mpp6 residues is conserved within eukaryotes, despite divergent sequence (*Figure 3A*). Within the helix, Mpp6 Tyr99 and Leu102 pack into a hydrophobic pocket created by Rrp40 Pro23 and Tyr56 (*Figure 3B*). While Mpp6 residues 101–103 are not conserved across evolution, Tyr99 is conserved as tyrosine, phenylalanine or leucine in other Mpp6 family members. Additional hydrophobic interactions are observed within the last beta strand of Mpp6, including contacts between Mpp6 Phe115 and Rrp40 Phe184

**Table 1.** Crystallographic data and refinement statistics.
One crystal was used. Highest resolution shell is shown in parenthesis.

| | Exo12 Dis3exo-endo-/Rrp6exo-/RNA/Mpp6Min |
|---|---|
| **Data collection** | |
| X-ray Source | APS GM/CA 23IDD |
| Space group | $P2_12_12_1$ |
| Cell dimensions | |
| a, b, c (Å) | 141.1, 213.6, 225.9 |
| α, β, γ (°) | 90.0, 90.0, 90.0 |
| Wavelength (Å) | 1.0332 |
| Resolution (Å) | 44.3–3.3 (3.42–3.3) * |
| $R_{merge}$ | 0.086 (0.613) |
| $I/\sigma I$ | 9.1 (1.7) |
| $CC_{1/2}$ | 0.997 (0.161) |
| Completeness (%) | 97.0 (95.0) |
| Redundancy | 3.4 (2.6) |
| Wilson B factor (Å$^2$) | 99.7 |
| **Refinement** | |
| Resolution (Å) | 44.3–3.3 |
| No. reflections observed | 341339 |
| No. unique reflections | 100440 |
| $R_{work}/R_{free}$ | 0.217/0.266 |
| No. atoms | 29498 |
| Protein | 29147 |
| RNA | 249 |
| Ligands | 38 |
| Water | 64 |
| Average B-factors | |
| Protein | 138 |
| RNA | 139 |
| Ligands | 141 |
| Water | 61 |
| R.m.s deviations | |
| Bond lengths (Å) | 0.001 |
| Bond angles (°) | 0.41 |
| Ramachandran plot | |
| % favored | 93.5 |
| % allowed | 6.5 |
| % outliers | 0 |
| Molprobity | |
| Clashscore/Percentile | 5.44/100[th] |
| MolProbity Score/Percentile | 1.72/100[th] |

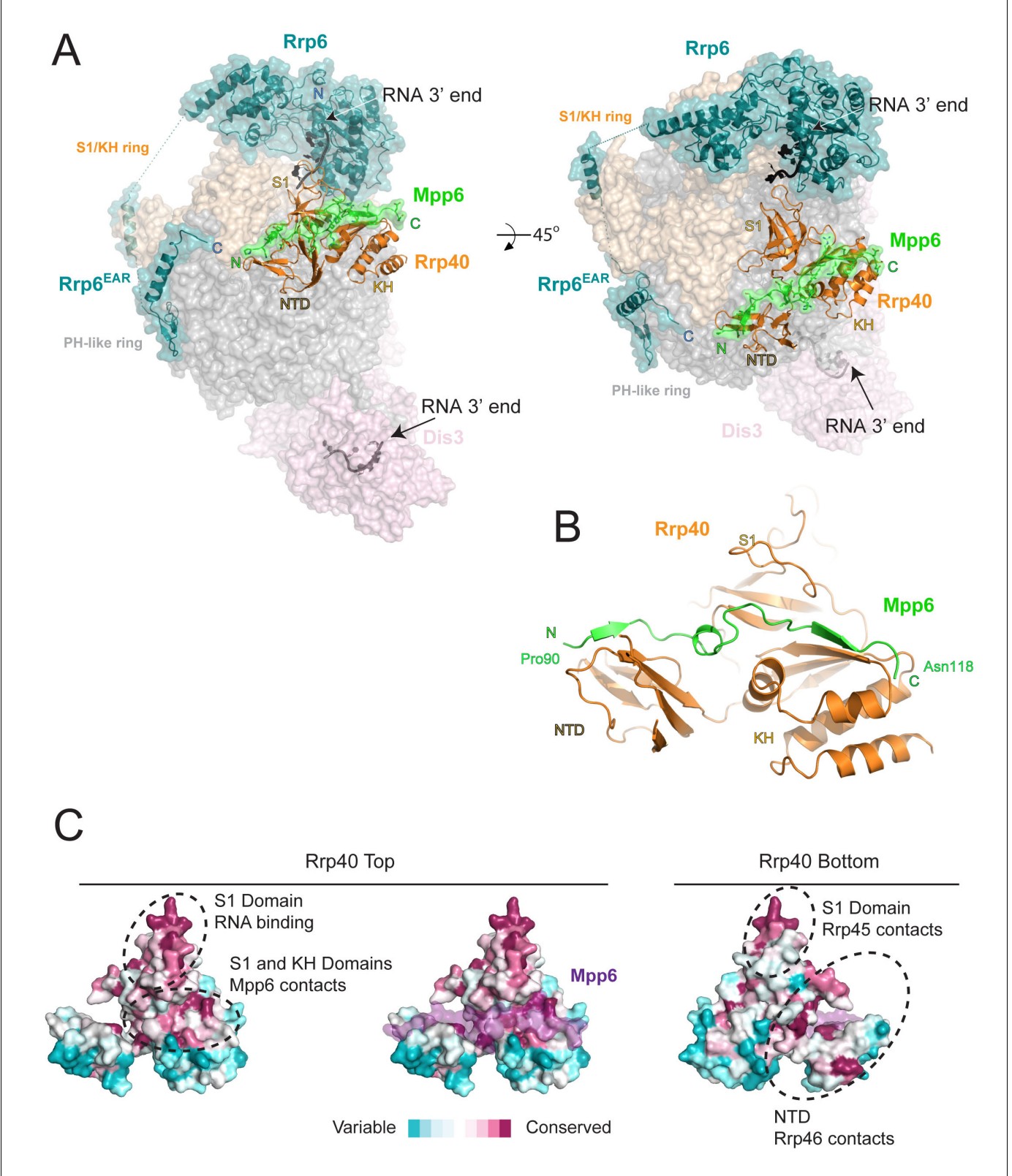

**Figure 2.** Structure of the 12-subunit Mpp6 nuclear exosome. (**A**) Global view of Exo12$^{Dis3exo-endo-/Rrp6exo-/Mpp6Min}$ bound to a 3'−3' RNA. Mpp6$^{Minimal}$ interacts with an extended surface across the S1/KH subunit, Rrp40. View from side (left) and top (right). Exosome subunits shown as surface view, Mpp6 in green, Rrp6 in teal, Rrp40 in cartoon (orange), RNA in black sticks. (**B**) Mpp6$^{Minimal}$ (green) makes extensive contacts to Rrp40 (orange) and spans all three of its domains. (**C**) Mpp6$^{Minimal}$ (transparent purple surface in middle and right panels) binds to a conserved surface of Rrp40. Other
*Figure 2 continued on next page*

*Figure 2 continued*

conserved surfaces important for RNA binding and scaffolding interactions to other exosome subunits are indicated. Surface conservation calculated with ConSurf (*Ashkenazy et al., 2010*).

The following figure supplement is available for figure 2:

**Figure supplement 1.** Rrp40 is necessary, but not sufficient for Mpp6 association with the exosome.

(*Figure 3B*). In yeast through man, the corresponding residue in Mpp6 is either a phenylalanine or tryptophan (*Figure 3A*), while a phenylalanine or leucine is present in Rrp40.

The most prominent electron density observed for Mpp6 corresponds to Arg112 within the last beta strand. Arg112 projects into a pocket formed between the S1 and KH domains of Rrp40 (*Figure 3B*; *Figure 2—figure supplement 1A*) and resides in one of two regions of Mpp6 previously identified as conserved from yeast to man (*Milligan et al., 2008*) (*Figure 3A*). Arg112 is within hydrogen bond distance to the Glu185 side chain carboxylate and the backbone carbonyl oxygen of Gly146 (*Figure 3B*), residues that are not known to be involved in other interactions and are strictly conserved in Rrp40. Given its location in the structure and evolutionary conservation, we refer to Mpp6 Arg112 as the 'arginine anchor'. Interestingly, two known mutations within Rrp40 that are associated with neurological disorders lie proximal to the arginine anchor of Mpp6. These include Gly148 (Gly191 in human), which is mutated to cysteine in hereditary spastic paraplegia in humans (*Halevy et al., 2014*), and Trp195 (Trp238 in human), which is mutated to arginine in pontocerebeller hypoplasia and spinal motor neuron degeneration (*Wan et al., 2012*). It is unclear how these mutations exert their phenotype or if they result in a loss of interaction with Mpp6.

To determine the importance of the arginine anchor, all four arginine residues present in Mpp6$^{\mathrm{Minimal}}$ were individually substituted to glutamate (R83E, R89E, R103E, R112E) and tested for their ability to stimulate Rrp6 in Exo10$^{\mathrm{Rrp6}}$ (*Figure 3C*). Three of the four Mpp6 mutants were defective in stimulating Exo10$^{\mathrm{Rrp6}}$ activity using polyA RNA, the exception being R83E, which is disordered in the structure. R112E exhibited the greatest defect, as would be predicted from removal of the arginine anchor (*Figure 3C*). The defect on Exo10$^{\mathrm{Rrp6}}$ stimulation observed for a second mutant, Mpp6 R89E, was exacerbated when combined with Rrp40 E185A, a mutation predicted to disrupt contacts to the Mpp6 arginine anchor (*Figure 3B*). The third Mpp6 mutant, R103E, exhibited a weaker defect that could be overcome by adding 10-fold excess mutant Mpp6. In contrast, the defect in exosome stimulation by Mpp6 R112E was not overcome by addition of excess cofactor (*Figure 3—figure supplement 1*). Combining Mpp6 R112E and Rrp40 E185A resulted in a similar loss of stimulation (*Figure 3C*). These results are consistent with contacts observed in the structure and suggest that Mpp6 association is required for stimulation of Rrp6 in the nuclear RNA exosome.

## Exosome stimulation by Mpp6 is dependent on the Rrp6 lasso

The first modeled residue of Mpp6$^{\mathrm{Minimal}}$ (Pro90) is located 13 Å away from the last modeled amino acids of the Rrp6 CTD (Phe618) and Rrp43 NTD (Thr9) as observed in previous structures of Rrp6-associated exosomes (*Figure 4A*) (*Makino et al., 2013, 2015*; *Wasmuth et al., 2014*; *Zinder et al., 2016*). The Rrp6 CTD is composed of two modalities: an Exosome Associating Region (EAR) that becomes ordered upon Exo9 association to tether Rrp6 above the S1/KH ring (*Makino et al., 2013*; *Wasmuth et al., 2014*), and the lasso, which includes the last one hundred residues and was recently reported to stimulate the activities of the nuclear exosome (*Wasmuth and Lima, 2017*) (*Figure 4B*). Unassigned electron densities were located between Mpp6, Rrp43 and Rrp6. To determine if Mpp6 contributed to these densities, anomalous data was collected from a crystal of Exo12$^{\mathrm{Dis3exo-endo-/Rrp6exo-/Mpp6Min}}$ containing selenomethionine substituted Rrp46/Rrp43. Anomalous peaks suggest that most unassigned densities likely correspond to a disordered loop in Rrp43 as Rrp43 Met323 is located proximal to this region (*Figure 4—figure supplement 1A*).

To determine if the Rrp6 lasso contributes to Mpp6-mediated stimulation of exosome activity, Exo10$^{\mathrm{Rrp6}}$ was reconstituted with an intact lasso (128–733), partial lasso (128–685) or lassoless Rrp6 (128–634) and assayed for decay of polyA RNA in the presence or absence of Mpp6 (*Figure 4C*). As stated earlier, lasso-intact Exo10$^{\mathrm{Rrp6}}$ can be stimulated up to 13-fold in the presence of Mpp6

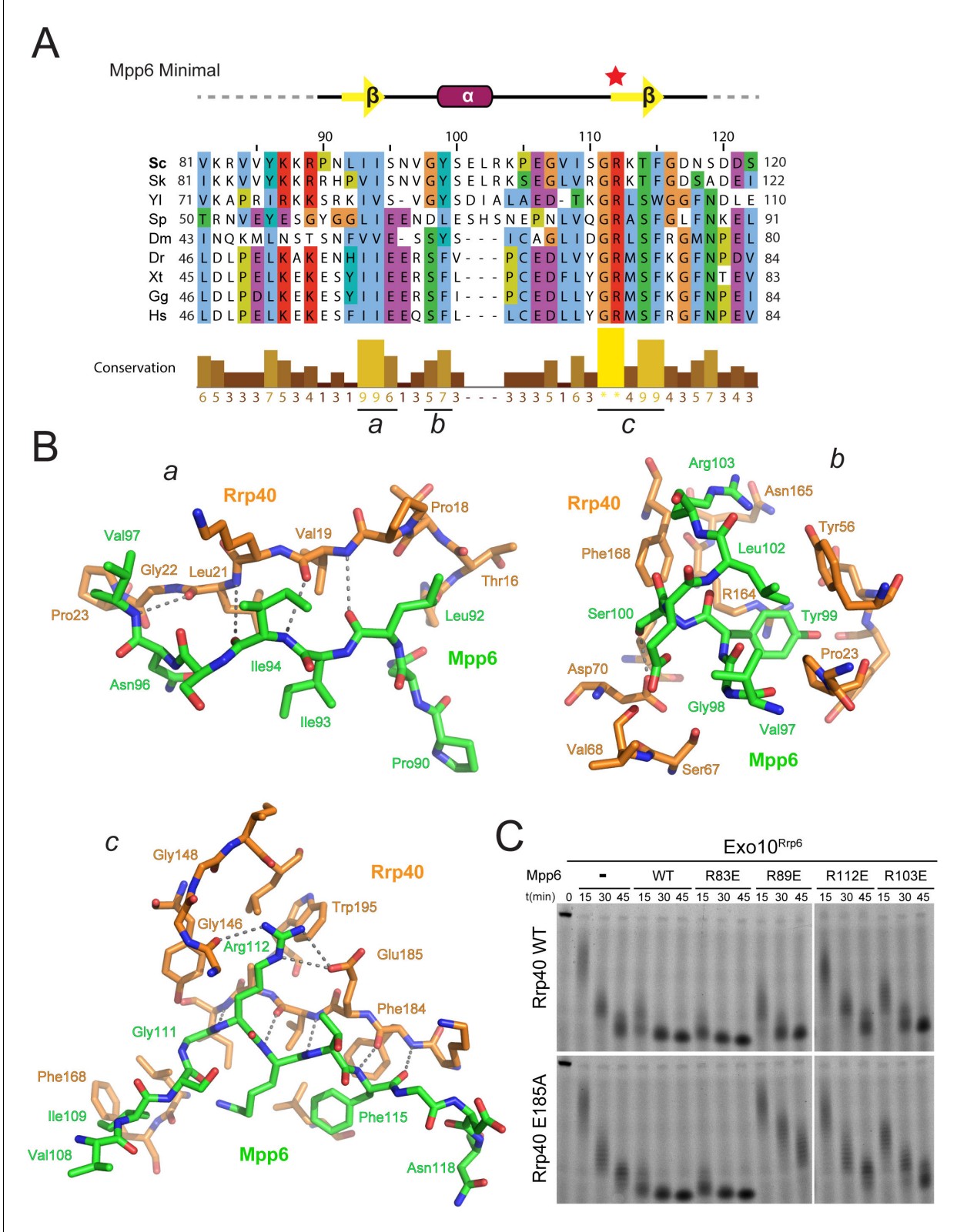

**Figure 3.** Conserved features of Mpp6 interaction with the S1/KH subunit, Rrp40. (**A**) Top: revised domain structure of Mpp6[Minimal] based on the crystal structure of Exo12[Dis3exo-endo-/Rrp6exo-/Mpp6Min]. Bottom: sequence alignment among *S. cerevisiae* Mpp6[Minimal] and other eukaryotes reveals conserved residues within Mpp6 [Minimal]. Sequences for *S.* kudriavzevii (*Sk*), *Y. lipolytica* (*Yl*), *S. pombe* (*Sp*), *D. melanogaster* (*Dm*), *D. rerio* (*Dr*), *X. tropicalis* (*Xt*), *G. gallus* (*Gg*), and *H. sapiens* (*Hs*). Sequence alignment and conservation calculated with Clustal Omega (*Larkin et al., 2007*) and Jalview

*Figure 3 continued on next page*

*Figure 3 continued*

(*Waterhouse et al., 2009*). Three regions (**a**, **b**, **c**) of high conservation are noted. (**B**) Stick representation of conserved contacts between Mpp6 (green) and Rrp40 (orange) are noted. Region *a* includes a hydrophobic patch that forms between the Rrp40 NTD and N-terminal residues of Mpp6$^{Minimal}$. Region *b* highlights an aromatic residue in Mpp6 that nestles between the NTD and KH domains of Rrp40. Region *c* focuses on the network of contacts between the 'arginine anchor' of Mpp6 and the S1 and KH domains of Rrp40. (**C**) Mpp6 arginine residues were individually substituted to glutamate residues to interrogate the importance of the arginine anchor. Representative RNA degradation assays of 5' fluorescein-labeled 49 nt polyA RNA and Exo10$^{Rrp6}$ with Mpp6$^{Minimal}$ WT and arginine mutants. Mutations disrupting the arginine anchor, Arg112, are most detrimental to Mpp6-mediated stimulation of Rrp6 activity. An additive effect is observed when a distal Mpp6 mutation, Arg89, is combined with a mutation in Rrp40 (E185A) that disrupts coordination of the Mpp6 arginine anchor.

The following figure supplement is available for figure 3:

**Figure supplement 1.** Conserved residues in Mpp6 interact with Rrp40.

(*Figure 1A*); however, either partial or complete removal of the Rrp6 lasso attenuates the magnitude of Mpp6-mediated stimulation of polyA RNA decay by Exo10$^{Rrp6}$ to 1.5 fold, suggesting that the Rrp6 lasso is important for stimulation by Mpp6. To determine if this trend holds for Mpp6-mediated stimulation of Dis3, Exo10$^{Dis3}$ was assayed with the Rrp6 CTD (EAR plus full lasso), Mpp6, or both. The Rrp6 CTD activated Exo10$^{Dis3}$ by 3-fold, consistent with previous observations that Dis3 exosomes are inactive on polyA RNA in the absence of intact Rrp6 (*Wasmuth and Lima, 2012*). While addition of Mpp6 to Exo10$^{Dis3}$ had no effect on Dis3 activity, combining the Rrp6 CTD and Mpp6 activated Exo10$^{Dis3}$ by 29-fold (*Figure 4D*), to levels indistinguishable from exosomes reconstituted with Rrp6$^{exo-}$ that contained an intact lasso (*Figure 4E*).

Finally, to determine if the Rrp6 EAR was required to position the lasso for Mpp6-mediated stimulation, exosomes were reconstituted with three different Rrp6$^{CAT}$-Csl4-Rrp6$^{Lasso}$ fusions that differ in lasso length and bypass the requirement of the Rrp6 EAR (*Figure 4—figure supplement 1B*) (*Wasmuth and Lima, 2017*) and assayed in the presence or absence of Mpp6. The Rrp6 lasso was necessary for Mpp6-dependent stimulation of exosome activity, regardless of whether the lasso was intact (residues 618–733) or composed of its distal half (residues 685–733) (*Figure 4—figure supplement 1C*). Collectively, these results suggest that Mpp6 cooperates with the Rrp6 lasso to stimulate the nuclear exosome.

## Comparison of Mpp6- and Rrp47-mediated stimulation of nuclear exosomes

Biochemical results thus far suggest that Mpp6 can stimulate the nuclear exosome. Previous biochemical studies have not revealed Rrp47-mediated stimulation of Rrp6 activity, but these studies used either truncated Rrp47 (*Schuch et al., 2014*), or truncated Rrp6 (*Dedic et al., 2014*; *Schuch et al., 2014*). To biochemically characterize the effects of Mpp6 and Rrp47 on nuclear exosome-mediated RNA decay, we repeated assays using full-length proteins given the recent reported relevance of the Rrp6 lasso (*Wasmuth and Lima, 2017*). Mpp6 or Rrp47 were mixed with Exo11$^{Dis3/Rrp6}$ individually or together and assayed for degradation of single-stranded polyA RNA (*Figure 5A*). Addition of Mpp6, Rrp47 or both resulted in 19-fold, 15-fold or 22-fold stimulation of Rrp6 activity on polyA RNA, respectively.

We next tested if Rrp47-mediated stimulation was also dependent on the Rrp6 lasso by assaying Exo11$^{Dis3/Rrp6}$ with Rrp6 containing intact and partial lassos. As before, Mpp6 stimulation of Rrp6 activity on polyA RNA was diminished by partial lasso deletion, however Rrp47 activity was not altered in this context (*Figure 5B*). This observation suggests that the stimulatory effects of Mpp6 and Rrp47 depend on unique elements within the nuclear exosome. These biochemical observations appear consistent with recent observations in yeast wherein deletion of the Rrp6 PMC2NT domain, a mutation known to result in loss of Rrp47 interaction (*Stead et al., 2007*), results in a synthetic growth defect when combined with deletion of the Rrp6 lasso (*Wasmuth and Lima, 2017*).

How might Mpp6 and Rrp47 stimulate Rrp6 activity in the nuclear exosome? RNA binding activities likely contribute, however alignment of Exo12$^{Dis3exo-endo-/Rrp6exo-/Mpp6Min}$ with Exo11$^{Dis3exo-endo-/Rrp6exo-}$ (*Zinder et al., 2016*) reveals changes in the orientation of the Rrp6 catalytic module (CAT) relative to Exo9 with Rrp6 CAT tilted by 15 degrees to a more open configuration in the complex

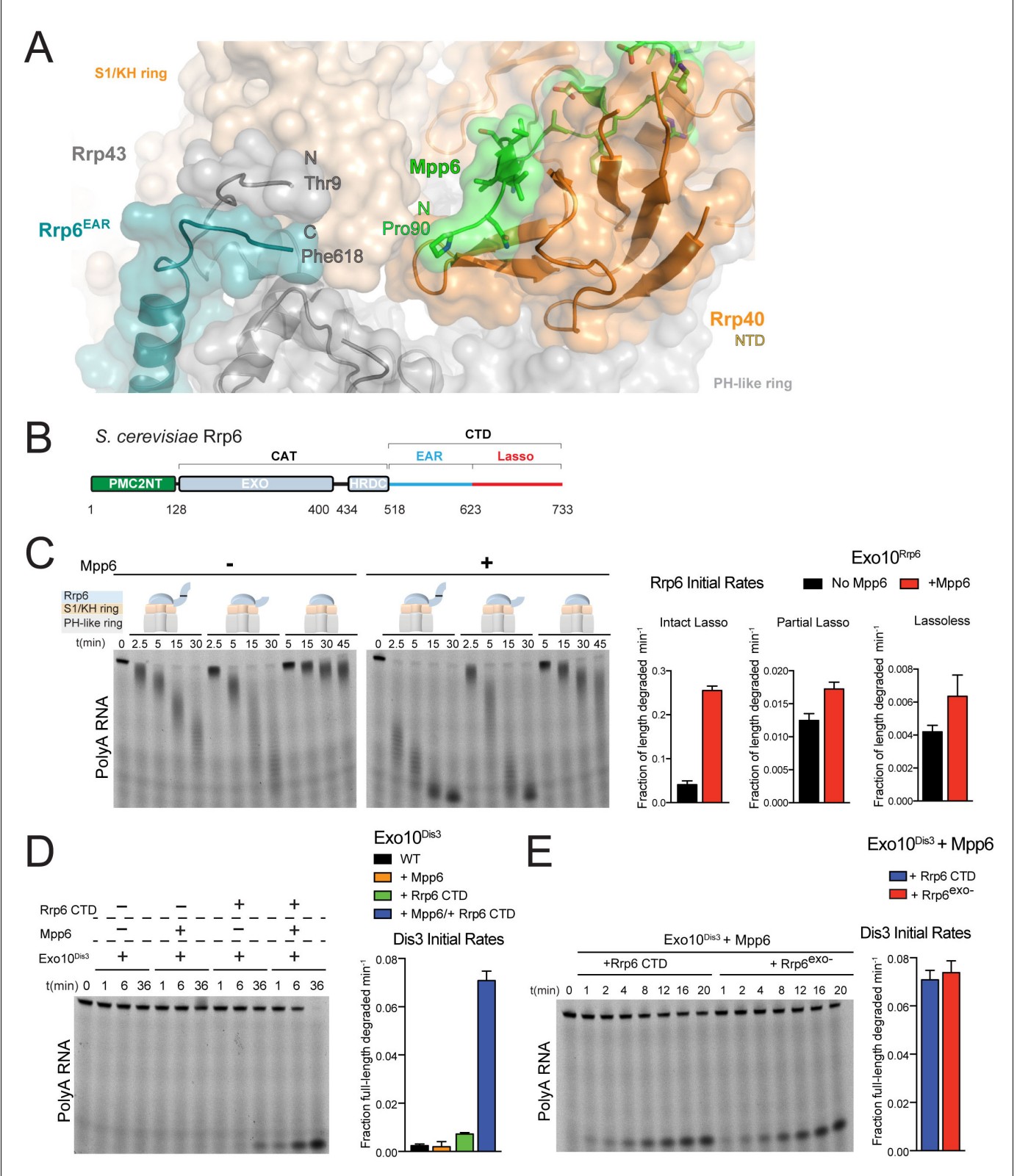

**Figure 4.** Mpp6 cooperates with the Rrp6 lasso to stimulate exosome activities. (**A**) Close-up view to show the C-terminal residue (Phe618) of the Rrp6 EAR (teal), the Rrp43 loop (residues 321–326 below Rrp6) and N-terminal residue (Thr9) (gray), and the N-terminal residue (Pro90) of Mpp6$^{Minimal}$ (green). Rrp6, Rrp43, Mpp6 and Rrp40 shown in cartoon and stick representation under a transparent surface. (**B**) Schematic domain structure of Rrp6. (**C**) Exo10$^{Rrp6}$ lacking part (128-685) or all of the Rrp6 lasso (128-634) are less stimulated by Mpp6 than Exo10$^{Rrp6}$ containing an intact lasso (128-733).

*Figure 4 continued on next page*

*Figure 4 continued*

Decay of 5'-fluorescein-labeled polyA 49 nt RNA. (**D**) Mpp6-dependent stimulation of Dis3 exoribonuclease activity in Exo10$^{Dis3}$ requires the Rrp6 CTD. Decay of 5'-fluorescein-labeled polyA 49 nt RNA. (**E**) The Rrp6 CTD stimulates Dis3 exoribonuclease activity in Exo10$^{Dis3}$ in a Mpp6-dependent manner similar to levels observed in the presence of Rrp6$^{exo-}$. Decay of 5'-fluorescein-labeled polyA 49 nt RNA. Bar graphs and error bars in panels **C**, **D**, and **E** are the result of triplicate experiments with error bars indicating plus or minus one standard deviation.

The following figure supplement is available for figure 4:

**Figure supplement 1.** Stimulation by Mpp6 depends on the Rrp6 lasso.

containing Mpp6. While lattice contacts between Rrp6 and two symmetry related Dis3 subunits may alter its conformation, Rrp6 is rotated away from the Exo9 central channel, perhaps facilitating greater access for RNA to engage the S1/KH ring and Rrp6 active site (*Figure 5—figure supplement 1A,B*). Interestingly, a similar conformation for Rrp6 was observed in the crystal structure of Exo12$^{Dis3exo-endo-/Rrp6exo-/Rrp47}$ (*Makino et al., 2015*), although in this case RNA was not observed in the Rrp6 active site. It remains unclear if Mpp6 and Rrp47 are responsible for the observed differences in Rrp6 conformations.

## Mpp6 can recruit Mtr4 to the exosome

Results presented thus far do not explain the synthetic lethality observed when Mpp6 is deleted along with Rrp6, or Air1, a component of the TRAMP complex (*Milligan et al., 2008*). Mutagenesis experiments designed to disrupt the structurally validated tripartite interaction between Rrp47-Rrp6 NTD-Mtr4$^{Nterm}$ did not alter yeast viability, Mtr4-exosome association, or RNA processing (*Schuch et al., 2014*). However, synthetic lethality was observed when mutations in the tripartite interaction with Mtr4 were combined with a Mtr4 C-terminal GFP tag, consistent with Mtr4 recruitment to the exosome being dependent on additional factors or Mtr4 surfaces that interact with Rrp6, Mpp6, Rrp47 or other components of the TRAMP complex. Furthermore, experiments in human suggest that both Mpp6 and Rrp47 contribute to Mtr4 recruitment (*Chen et al., 2001*; *Schilders et al., 2007*; *Lubas et al., 2011*).

Mpp6 and/or Rrp47 were added to full-length Exo11$^{Dis3/Rrp6}$ to determine if these cofactors contribute to interactions with Mtr4 via analytical gel filtration (*Figure 5—figure supplement 1C*). Consistent with previous results, Mtr4 did not co-elute with the exosome in the absence of Rrp47 and Mpp6 while co-elution was observed in the presence of Rrp47 (*Schuch et al., 2014*) (*Figure 5C* and *Figure 5—figure supplement 1C*). When Mpp6 and Rrp47 were both present, more Mtr4 was detected in fractions containing the exosome suggesting greater stability of the complex. Importantly, Mpp6 was capable of recruiting Mtr4 to the exosome in the absence of Rrp47 and, to lesser extent, in the absence of Rrp6. Removing the C-terminal 30 amino acids of Mpp6 (1-156) did not disrupt Mtr4 interaction, but removal of the N-terminal 81 amino acids in Mpp6 (82-186) or the conserved N-terminal 22 amino acid motif in Mpp6 (23-120) resulted in no detectable interactions with Mtr4. Mpp6$^{Minimal}$ (81-120) failed to interact with Mtr4. These data suggest that Mpp6 elements required for exosome activation are distinct from those required for Mtr4 recruitment.

To assess whether Mpp6- or Rrp47-mediated recruitment of Mtr4 results in Mtr4-dependent RNA degradation, we assayed reconstituted exosomes for decay activity in the presence of exogenously added Mtr4 using a 17 nt double stranded RNA with a 10 nt 3' poly(A) overhang (ds$_{17}$A$_{10}$) (*Schuch et al., 2014*) in comparison with assays using the same labeled 27 nt RNA without the complementary 17 nt strand (ss$_{17}$A$_{10}$) (*Figure 6A*). Previous studies reported that Dis3 is unable to degrade the ds$_{17}$A$_{10}$ substrate in the nuclear exosome (*Makino et al., 2015*), presumably because the 3' overhang is too short to span the Exo9 central channel. Rrp6-dependent trimming of the 3' overhang was stimulated in the presence of Rrp47, Mpp6 or both (*Figure 6*), consistent with our biochemical observations using polyA RNA (*Figure 5*). Products of Dis3 activity could be detected with the ds$_{17}$A$_{10}$ substrate, but only when Rrp47 or Mpp6 was present. Importantly, this activity was dependent on the presence of both Mtr4 and ATP, as AMPPNP, a non-hydrolyzable ATP analog, failed to support activity. By contrast, Dis3-dependent degradation of ss$_{17}$A$_{10}$ did not appear dependent on Mpp6, Rrp47, Mtr4, ATP or AMPPNP. Trimming of the ds$_{17}$A$_{10}$ 3' overhang by Rrp6 dominated the assay in WT exosomes in the presence of Rrp47 or Mpp6, so we analyzed Mtr4-

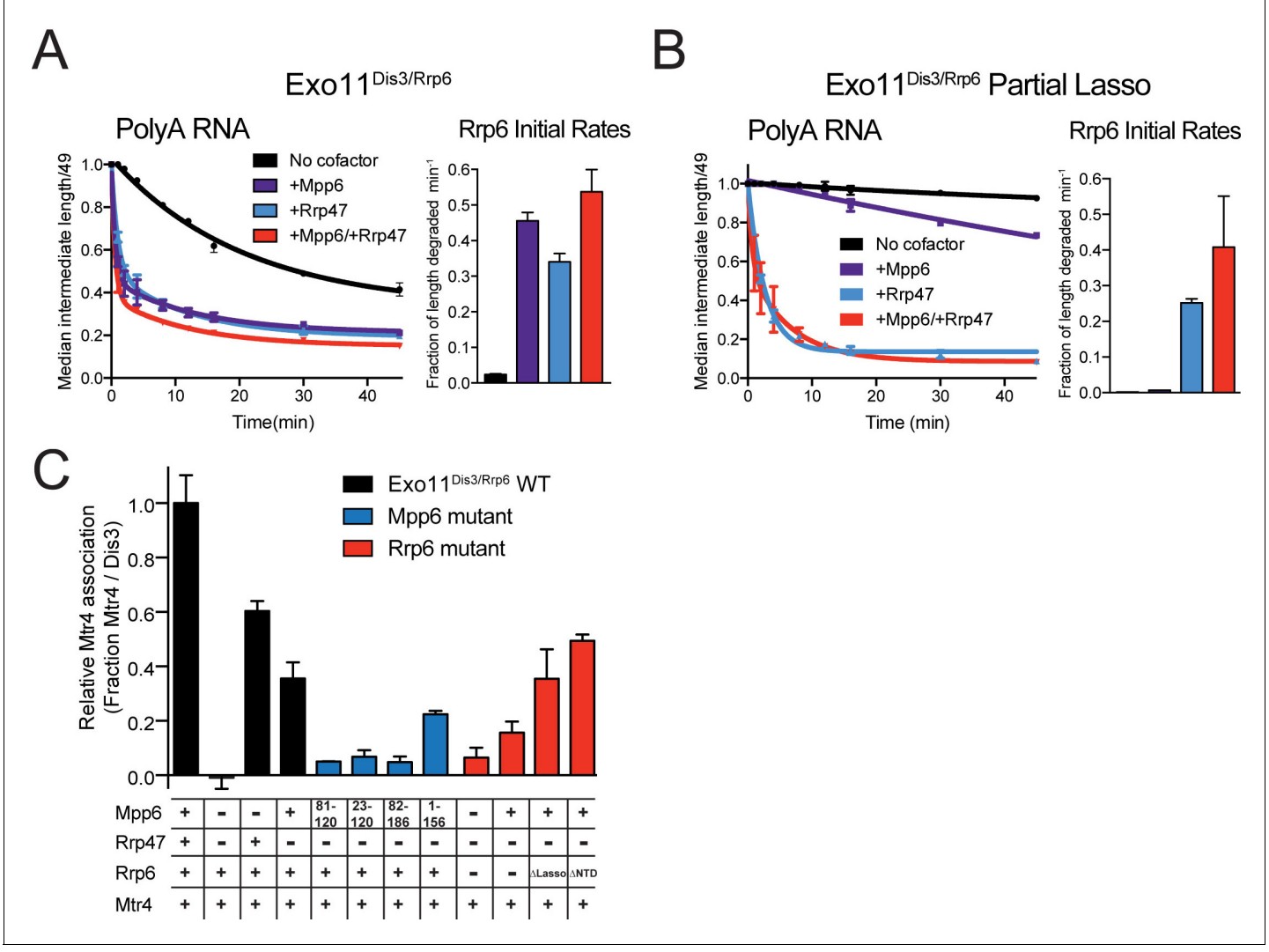

**Figure 5.** Mpp6 and Rrp47 stimulate Rrp6 activity and contribute to Mtr4 recruitment. (**A**) Both Mpp6 and Rrp47 stimulate Rrp6 activity in degradation of a single-stranded 5' fluorescein-labeled 49 nt polyA RNA in Exo11$^{Dis3/Rrp6}$. Initial rates from triplicate experiments shown, with error bars representing plus or minus one standard deviation. (**B**) Mpp6, but not Rrp47, requires the Rrp6 lasso to stimulate Rrp6 activity on 5' fluorescein-labeled 49 nt polyA RNA. Initial rates from triplicate experiments are shown, with error bars representing plus or minus one standard deviation. (**C**) Mpp6 contributes to Mtr4 recruitment in Exo11$^{Dis3/Rrp6}$. Analytical gel filtration experiments were performed with Exo10$^{Dis3}$ and Exo11$^{Dis3/Rrp6}$, Rrp47, Mtr4, with various truncations of Mpp6 and Rrp6 as indicated. Bar graphs represent the ratio of Mtr4 (122 kDa) to Dis3 (114 kDa) in peak fractions of the complex (***Figure 5—figure supplement 1C***), as calculated by densitometric analysis of the fractions on SDS-PAGE, with error bars representing plus or minus one standard deviation.

The following figure supplement is available for figure 5:

**Figure supplement 1.** Superposition of Exo12$^{Dis3exo-endo-/Rrp6exo-/Mpp6Min}$ onto Exo11$^{Dis3exo-endo-/Rrp6exo-}$ (PDB 5K36) and Mtr4 interactions with exosomes as analyzed by gel filtration and SDS-PAGE.

dependent decay using exosomes reconstituted with catalytically inactivated Rrp6 (Rrp6$^{exo-}$). As before, Mtr4-dependent decay was only observed in the presence of Rrp47 or Mpp6, while addition of both cofactors resulted in the greatest stimulation (***Figure 6A***). These results show that Mtr4 can promote degradation of dsRNA by the nuclear exosome, and that this activity is dependent on Mpp6 and Rrp47 cofactors.

Unwinding of the dsRNA region by Mtr4 appears responsible for Dis3 activity, but only when it is recruited to the exosome via Rrp47 or Mpp6. To determine the relevance of domains in Mpp6

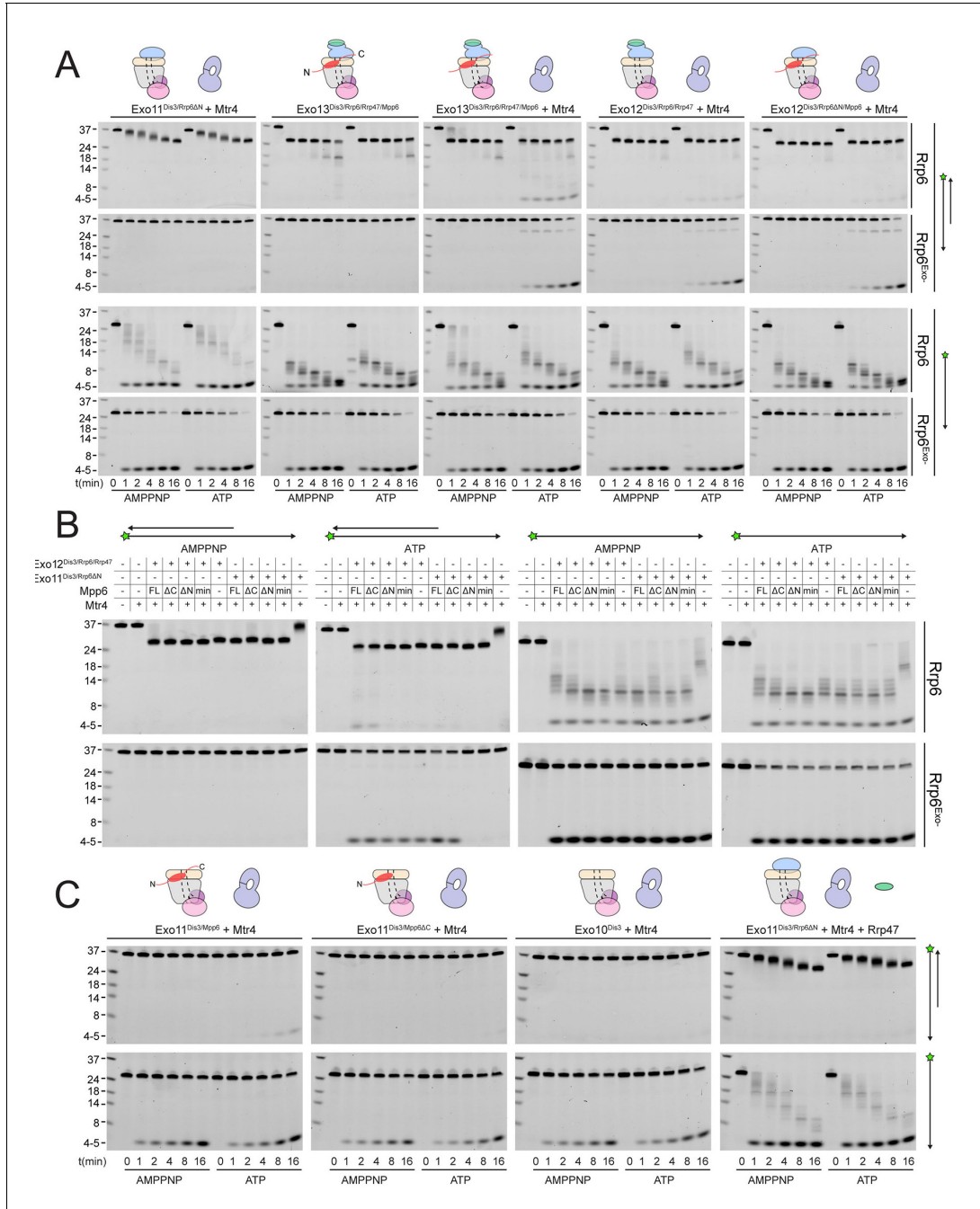

**Figure 6.** Mtr4-dependent RNA degradation requires either Mpp6 or Rrp47. Urea-PAGE analysis of RNA decay products by indicated exosome complexes and cofactors. In reactions labeled 'Rrp6$^{Exo-}$', Rrp6 contains a D238N mutation to render its exonuclease site catalytically inactive. 5' fluorescein-labeled polyA RNA of indicated lengths are present in the leftmost lane of each gel with the Dis3 product labeled '4–5'. Samples were not heated prior to gel electrophoresis, so the dsRNA substrate runs as a duplex at approximately 37 nt (**A**) Activity of reconstituted RNA exosome complexes in the presence or absence of equimolar exogenous Mtr4. (**B**) Mix-in of 1.5-fold molar excess of full-length Mpp6 or indicated truncations in RNA decay reactions containing reconstituted Exo12$^{Dis3/Rrp6/Rrp47}$ or Exo11$^{Dis3/Rrp6ΔN}$ and equimolar Mtr4 (when present). Results shown for an end point assay after 1 min incubation with exosomes containing Rrp6 or 8 min for exosomes containing Rrp6$^{exo-}$. (**C**) RNA degradation activities of reconstituted RNA exosome complexes and indicated mutations in the presence of equimolar exogenous Mtr4 and/or Rrp47. Cartoons shown above gels in panels A and C depict the exosome core in grey and wheat, Rrp6 in blue, Dis3 in pink, Mpp6 in red and Rrp47 in green. The central body of Mpp6 is shown as an ellipse with N- and C-terminal tails labeled or removed to reflect the protein used in the assay. The N-terminal PMC2NT domain of Rrp6 is shown as an appendage to the Rrp6 protein.

required for this activity, we compared activities of exosomes in the presence of exogenous Mpp6 (FL: 1–186, ΔC: 1–120, ΔN: 81–186, and Min: 81–120) to those lacking Mpp6 in the presence of Mtr4. Results suggest that each Mpp6 construct stimulates Rrp6 trimming activity of the 3' overhang, but only FL and ΔC Mpp6 constructs, both of which recruit Mtr4 in gel filtration experiments (*Figure 5C*), support ATP/Mtr4/Dis3-dependent RNA decay in the absence of Rrp47 (*Figure 6B*). In the presence of Rrp47, Mpp6 is largely dispensable for this activity, but FL and ΔC Mpp6 further increased Mtr4-dependent decay. As before, similar trends were observed in assays conducted with exosomes containing Rrp6$^{exo-}$.

Perhaps consistent with the observation that yeast strains are viable in the absence of Rrp6 and Rrp47 (*Feigenbutz et al., 2013b*; *Stuparevic et al., 2013*), some Mtr4-dependent decay could be observed in complexes lacking Rrp6 (*Figure 6C*), although this required full-length Mpp6 and its activities were weaker in comparison to exosomes containing Rrp6 or Rrp6 plus Rrp47. Furthermore, addition of Mpp6 to exosomes lacking Rrp6 or Rrp47 also showed an ability to interact with Mtr4 (*Figure 5C*). These data suggest that Mpp6 could contribute to Mtr4 recruitment in the absence of Rrp6 and Rrp47 proteins (*Schuch et al., 2014*). Importantly, ATP/Mtr4/Dis3-dependent activity on the ds$_{17}$A$_{10}$ substrate was only observed in complexes that could physically interact with Mtr4 (*Figure 5C*), suggesting that the unwinding activities of Mtr4 lead to productive degradation only when Mtr4 is physically tethered to the RNA exosome.

## Optimal cell growth depends on unique domains in Rrp6 and Mpp6

Genetic interactions between Mpp6, Rrp47 and Rrp6 have been reported previously (*Feigenbutz et al., 2013a*; *Milligan et al., 2008*). While none of these genes is essential, deletion of Mpp6 and Rrp6 results in synthetic lethality. Furthermore, viability is maintained in cells lacking both Rrp47 and Rrp6 as long as Mpp6 remains present. To relate the relevance of Mpp6 and Rrp6 domains defined in this study as important for exosome stimulation and Mtr4 recruitment to function in vivo, strains lacking Rrp6 and Mpp6 were complemented with mutant alleles for Rrp6 and/or Mpp6 in various combinations (*Figure 7*, *Figure 7—figure supplement 1*). As stated previously, strains lacking both Rrp6 and Mpp6 are not viable while strains containing either full-length Mpp6 or full-length Rrp6 are viable. Importantly, in strains lacking Mpp6 removal of the PMC2NT domain in *rrp6 (128-733)*, a combination that should eliminate Rrp6 interactions with Rrp47, and by extension the Mtr4 helicase, resulted in slower growth while removal of both the Rrp6 PMC2NT and lasso in *rrp6 (128-634)* resulted in no viable colonies (*Figure 7B*). In contrast, in strains lacking Rrp6, only full-length Mpp6 suppressed lethality as strains complemented with Mpp6 truncations *mpp6 (1-120)*, *mpp6 (81-186)* and *mpp6 (81-120)* were not viable (*Figure 7B*, *Figure 7—figure supplement 1*). This observation is consistent with our in vitro data, namely that in the absence of Rrp6, full-length Mpp6 can still mediate recruitment of Mtr4 to the exosome (*Figure 5C*; *Figure 5—figure supplement 1C*). In addition, growth defects associated with *rrp6 (128-733)*, an allele that should not interact with Rrp47, are suppressed by WT Mpp6 and *mpp6 (1-120)*, but not other *mpp6* variants. This result is perhaps consistent with biochemical observations, namely that Mpp6 residues 1–120 contribute to recruitment of Mtr4 and Mtr4-dependent decay in the absence of Rrp47 (*Figure 5C*; *Figure 6*).

Genetic data also appear consistent with biochemical observations that Mpp6 cooperates with the Rrp6 lasso for function. While *mpp6 (1-120)* suppressed growth defects observed for *rrp6 (128-733)*, it did not support growth when combined with the lassoless allele of *rrp6 (128-634)*. Strains containing *mpp6 (81-120)* phenocopied Δmpp6, suggesting that this construct is not expressed or that the exosome association domain is not sufficient to support Mpp6 functions in vivo, despite its ability to stimulate nuclear exosome activity in vitro (*Figure 1C*). However, since the Mpp6 (81-120) protein did not result in productive interactions with Mtr4 in vitro (*Figure 5C*; *Figure 6*), the observation that *mpp6 (81-120)* phenocopies Δmpp6 is also consistent with a model in which Mtr4 recruitment is defective. Finally, a strain harboring *mpp6 (81-186)* and *rrp6 (128-634)* was viable despite the fact that both Rrp47- and Mpp6-dependent mechanisms for Mtr4 recruitment are presumed missing. Although this strain grows much slower than any other characterized in this study, its viability suggests that combining the Mpp6 exosome associating region with its C-terminal domain is sufficient to suppress lethality observed for *rrp6 (128-634)*, perhaps through interactions with the transcriptional termination machinery via Nrd1 as noted previously (*Kim et al., 2016*).

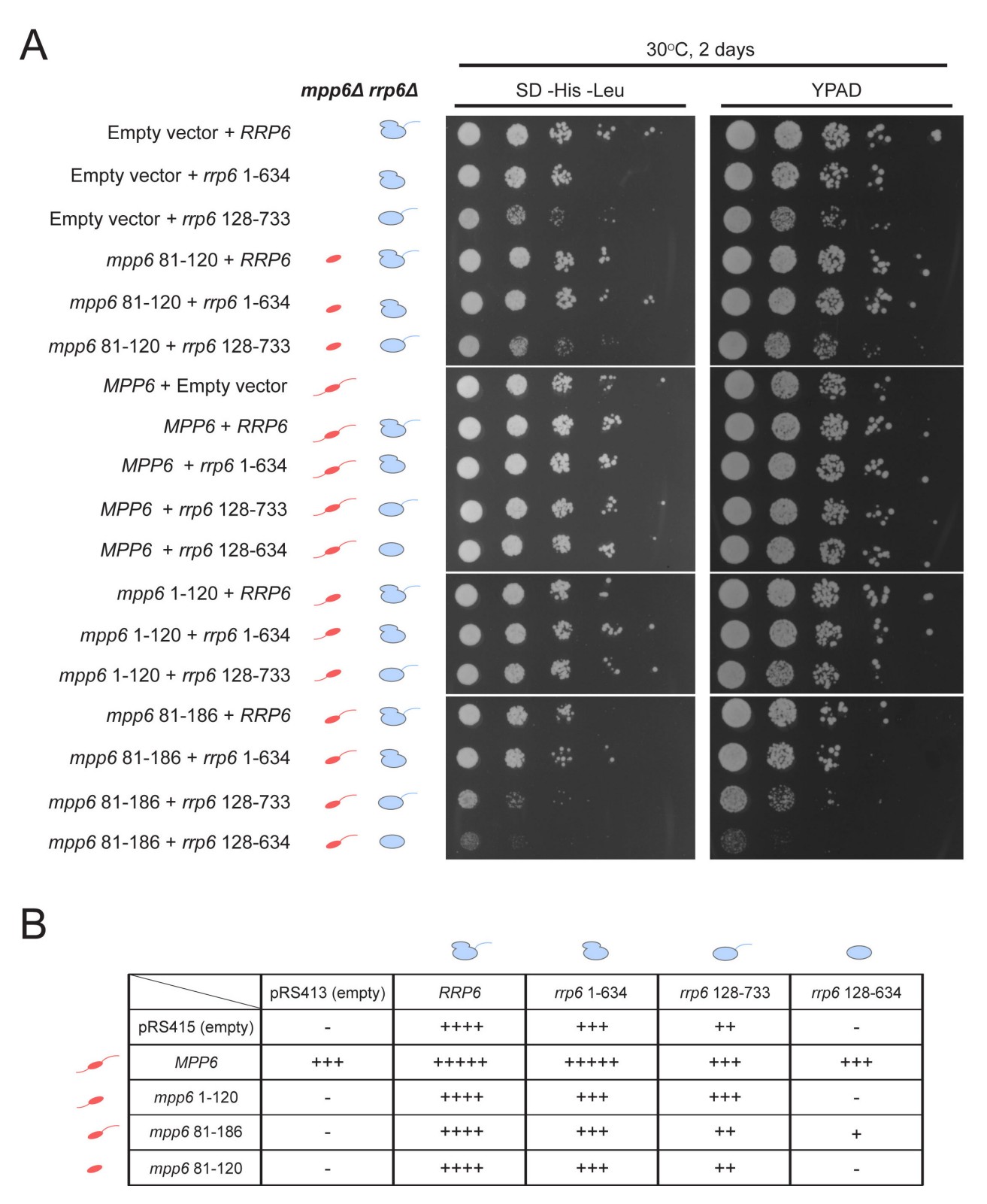

**Figure 7.** Optimal cell growth depends on unique domains in Rrp6 and Mpp6. (**A**) Growth analysis of *Saccharomyces cerevisiae* strains carrying viable combinations of Rrp6 and Mpp6 alleles. Ten-fold dilutions series of the *rrp6Δ mpp6Δ* strains transformed with the indicated pRS415 *mpp6* and pRS413 *rrp6* plasmids. Spotting was performed on SD-His-Leu or YPAD solid media and cells were incubated at 30°C for 2 days. Cartoons depict *mpp6* and *rrp6* alleles with respect to N- and C-terminal deletions. (**B**) Scoring table of the yeast growth phenotypes established in panel A and *Figure 7—figure*
*Figure 7 continued on next page*

Figure 7 continued

supplement 1B after 1 day growth. Scoring is based on a subjective five-point system where five '+' symbols ('+++++') correspond to the fastest observed growth rate and one '+' symbol to the slowest growth rate. Plasmid combinations indicated with the '- 'symbol resulted in synthetic lethality (see *Figure 7—figure supplement 1A*).

The following figure supplement is available for figure 7:

**Figure supplement 1.** Selection of viable yeast strains carrying different combinations of pRS415 *mpp6* and pRS413 *rrp6* plasmids.

## Discussion

Experiments presented here suggest that the nuclear exosome cofactor, Mpp6, can associate with the Exo9 core through interactions with the S1/KH subunit Rrp40 to recruit the helicase Mtr4 and to stimulate the activities of the exosome, most notably those catalyzed by Rrp6. A minimal region of Mpp6 comprised of amino acids 81–120 appears sufficient for exosome association and stimulation in vitro, although it is defective for recruitment of Mtr4. As our structure shows, this fragment includes a conserved arginine anchor that is important for Mpp6 interactions with the exosome core via Rrp40 and biochemical interactions with Rrp6 and the Rrp6 lasso.

This study also sheds light on dual functions of Mpp6 and Rrp47 insofar as both factors can stimulate RNA exosome activities and both contribute to Mtr4 recruitment and Mtr4-dependent degradation in vitro, perhaps providing some additional insights to in vivo synthetic interactions reported in previous studies (*Feigenbutz et al., 2013a*; *Milligan et al., 2008*). Interactions between the nuclear exosome and Rrp47 or Mpp6 result in stimulation of exosome activities, but they rely on distinct surfaces for these activities. Rrp47 interacts with the N-terminal PMC2NT domain of Rrp6 (*Makino et al., 2015*; *Schuch et al., 2014*), while Mpp6-mediated stimulation requires the Exo9 core and the Rrp6 C-terminal lasso.

Rrp47 and Mpp6 each contribute to Mtr4 interactions with Exo11$^{Dis3/Rrp6}$, and inclusion of both cofactors appears to maximize these interactions. While we were unable to decouple Rrp47-mediated stimulation from its ability to recruit Mtr4, genetic and biochemical results suggest that a minimal fragment of Mpp6 (residues 81–120) that supports Mpp6-mediated stimulation was not sufficient to support growth in vivo when paired with an *rrp6* allele that does not interact with Rrp47, and that additional Mpp6 elements are required for optimal cell growth. These include N-terminal elements that contribute to interactions with Mtr4 and C-terminal elements that were reported to contribute to interactions with the termination apparatus or perhaps Rrp6 as indicated by two-hybrid studies (*Kim et al., 2016*). Additional details have been resolved with respect to the contributions of Mpp6 and Rrp47 to nuclear exosome function, however critical work remains to resolve the structural and mechanistic bases for activities associated with the Mtr4 helicase.

## Materials and methods

### Exosome subunit purification, complex reconstitution, and analyses

Cloning, expression, purification of exosome subunits and reconstitution of various complexes have been described previously (*Greimann and Lima, 2008*; *Wasmuth and Lima, 2012*; *Wasmuth et al., 2014*; *Zinder et al., 2016*). *S. cerevisiae* Mpp6 (full-length and various truncations), Rrp47, and Mtr4 were cloned into pRSF-Duet1 with a N-terminal Smt3-fusion tag, and were transformed into *E. coli* BL21 (DE3) RIL (Novagen; St. Louis, MO). Recombinant protein expression was induced by addition of 0.4 mM isopropyl-$\beta$-D-thiogalactoside and overnight shaking at 18°C. Cells were lysed by sonication, and supernatants were purified by Ni-NTA. For Mpp6 variants and Rrp47, further purification was performed on a Superdex 75 (GE; Pittsburgh, PA), followed by overnight cleavage of the Smt3 tag using Ulp1 protease, and a final purification on a heparin Hi-Trap column to remove Smt3 and nucleic acid impurities. Mtr4 was purified on a Superdex 200 (GE), subject to Ulp1 cleavage overnight, and purified by MonoQ (GE) to remove Smt3. To generate selenomethionine-bearing Mpp6$^{Minimal}$, PCR-based site-directed mutagenesis was used to generate I94M to introduce a single methionine in this construct. Selenomethionine-bearing Smt3-Mpp6$^{Minimal}$ and Smt3-Rrp46/Rrp43

were each grown in 4L Lemaster medium and induced at 18°C overnight. Proteins were purified and reconstituted as their native variants.

For interactions with Mpp6, 1.7 µM Exo9 was incubated with 5-fold molar excess Mpp6 on ice for 30 min in a total volume of 400 microliters of 100 mM NaCl, 20 mM Tris pH8.0, 1 mM TCEP, and run on a Superose 6 (GE), and fractions analyzed by SDS-PAGE and stained by Sypro Ruby. For interactions with Mtr4, 1.0 µM Exo11$^{Dis3/Rrp6}$ was incubated with 5-fold molar excess Mpp6 or Rrp47, and 2-fold molar excess Mtr4 on ice for 30 min in a total volume of 400 microliters of 100 mM NaCl, 40 mM MES pH6.5, 1 mM TCEP and run on a Superdex 200 Increase (GE), and fractions analyzed by SDS-PAGE and stained with Sypro Ruby. Densitometry was used to calculate the ratios of Mtr4 to Dis3 in fractions corresponding to nuclear exosome peaks using Fujifilm Multi Gauge.

## 3′−3′ RNA synthesis and purification

Alkynyl (5′ hexynyl UUU AUU AUU UAU UUU AAA A 3′) and azido (5′ azide/NHS UUA UUU UAA AA 3′) RNAs were synthesized and purified by Integrated DNA Technologies (Coralville, IA). RNAs were ligated by incubating 200 µM alkynyl RNA with 100 µM azido RNA in 80 mM potassium phosphate (pH 7.0), 5 mM sodium ascorbate, 0.5 mM CuSO$_4$/THPTA at 25°C for 1 hr. Quenching and purification of the reaction by DEAE chromatography were performed as previously described (*Zinder et al., 2016*).

## Crystallization and structure determination

Native and selenomethionine bearing Exo12$^{Dis3exo-endo-/Rrp6exo-/Mpp6}$ were mixed in a 1:1 molar ratio with a single-stranded 30 nucleotide RNA bearing two 3′ ends synthesized by click chemistry (*Zinder et al., 2016*) and allowed to incubate on ice for 30 min. After one week, the protein-RNA complex formed single crystals at 4°C in 11–13% PEG3350, 100 mM NaCitrate pH 5.6, 7 mM MES pH 6.5, and 175–200 mM ammonium sulfate that continued to grow for up to one month. Crystals were cryoprotected in mother liquor augmented with 21% glycerol, introduced in three steps of 7% increments. Native and anomalous x-ray diffraction data were collected at the Advanced Photon Source 24-ID-E and 24-ID-C beam lines (NECAT) (RRID:SCR_008999), and 23-ID-D (GM/CA) at the selenium edge. Data were processed using HKL2000 (*Otwinowski and Minor, 1997*) and the structure solved by molecular replacement using Phaser (*McCoy et al., 2007*) (RRID:SCR_014219) in the Phenix suite (*Adams et al., 2010*) (RRID:SCR_014224) with the coordinates of yeast Exo10$^{Rrp6exo-}$ (PDB: 4OO1) and Exo11$^{Dis3exo-endo-/Rrp6exo-}$ (PDB: 5K36) as search models (*Table 1*). Anomalous maps were calculated and iterative rounds of refinement were accomplished using Phenix. RNA and protein were manually built using Coot (*Emsley et al., 2010*) (RRID:SCR_014222). Simulated annealing omit maps and maps used during building were also generated using CNS (*Brunger, 2007*) (RRID:SCR_014223). The model was refined using positional refinement, real-space refinement and individual B-factor refinement. Figures depicting structures were prepared with PyMol (Schrödinger) (RRID:SCR_000305). Surface conservation was calculated using ConSurf (*Ashkenazy et al., 2010*) (RRID:SCR_002320). Structure quality was assessed using MolProbity (*Chen et al., 2010*) (RRID:SCR_014226).

## RNA degradation assays

RNA oligonucleotides were synthesized with a 5′ fluorescein as described previously (*Liu et al., 2006*; *Wasmuth et al., 2014*) and purchased from Integrated DNA Technologies (IDT) or Dharmacon (Lafayette, CO). With the exception of *Figures 5* and *6*, all exoribonuclease assays used 1 nM enzyme with 10 nM 5′-fluorescein-labeled RNA. For the exoribonuclease assays in *Figure 5*, 2.5 nM enzyme was assayed with 10 nM 5′-fluorescein-labeled 49 nt polyA RNA. For experiments requiring mix-ins, 3-fold molar excess cofactor was added to reconstituted exosome and incubated on ice for 30 min before initiating decay. All exoribonuclease assays were conducted using in 50 mM KCl, 20 mM Tris-HCl (pH8.0), 10 mM DTT, 0.5 mM MgCl$_2$, and 1 U/µL RNase inhibitor (New England Biolabs; Ipswich, MA) at 30°C, as described previously (*Wasmuth and Lima, 2012*). RNA intermediates were resolved by denaturing PAGE and visualized with a Fuji FLA-5000 fluoroimager. Quantification of ribonuclease activity was performed as described previously (*Wasmuth and Lima, 2012*). Briefly, to quantify Dis3 exoribonuclease activity, the fraction of full-length RNA degraded at a given time was calculated based on the amount of 4–5 nt products released. Rrp6 exoribonuclease activity

was quantified by dividing the median length of the RNA intermediates at a given time by the full-length RNA substrate. This was then used to calculate the fraction of RNA length degraded per minute using the equation ((1 – median length products/substrate length)/min). For quantitation of Rrp6 and Dis3 activities when present in the same complex, the rate of accumulation of Dis3 products (4–5 nt) was analyzed at time points before Rrp6 products merged with those of Dis3. For both activities, data from triplicate experiments were analyzed using GraphPad Prism 6 (RRID:SCR_002798), and initial rates were calculated from data obtained within the linear range. Endoribonuclease assays utilized 10 nM exoribonuclease dead exosomes and 10 nM 5' fluorescein-labeled RNA, and were initiated with 3 mM $MnCl_2$. The Rrp6 protein used in RNA degradation assays in *Figures 1*, *2*, *3* and *4* included an N-terminal deletion of the PMC2NT domain to prevent aggregation (128-733) or respective C-terminal truncations. Assays presented in *Figures 5* and *6* utilized full-length Rrp6 or indicated truncations to include the PMC2NT domain that is required for interaction with Rrp47.

The 5' 6-carboxyfluorescein bottom strand (5' FAM CCC CAC CAC CAU CAC UUA AAA AAA AAA 3') and duplex top strand (5' AAG UGA UGG UGG UGG GG 3') were synthesized and HPLC purified by IDT. To prepare the $ds_{17}A_{10}$ substrate used in *Figure 6*, top and bottom strands were mixed at 100 and 150 μM, respectively, in 20 μL of 10 mM Tris-Cl pH 8.0, 100 mM KCl, 0.5 mM EDTA, heated to 95°C for 5 min, cooled to 60°C for 2 min, then to 4°C in a thermocycler. The $ss_{17}A_{10}$ was prepared in a similar manner, except the top strand was excluded from the reaction. Annealing reactions were fractionated by DEAE chromatography (Waters Protein-Pak DEAE 8 HR 1000 Å 8 μm 5 × 50 mm column) using a linear gradient from 300 mM NaCl to 700 mM NaCl in 10 mM Tris-Cl pH 8.0, 0.1 mM EDTA pH 8.0 at 40°C. Fractions were collected and analyzed by TBE-PAGE using SYBR Gold staining (Life Technologies; Carlsbad, CA). RNA in fractions containing the desired product was ethanol precipitated and the pellet resuspended in ultrapure $H_2O$.

For RNA degradation assays in *Figure 6*, the final concentrations were 10 nM exosome, 10 nM Mtr4 if present, 1 mM ATP or AMPPNP, 0.5 U/μL RNAse inhibitor (New England Biolabs), and 10 nM RNA substrate in 20 mM HEPES-KOH pH 7.5, 50 mM KCl, 1.1 mM $MgCl_2$, 0.5 mM TCEP-KOH pH 7.0, 0.02% IGEPAL co-630. 50 mM ATP and AMPPNP stock solutions were adjusted to pH 7.0 with KOH. Exosome complexes were incubated on ice with Mtr4 at 1 μM each in the above buffer for 1 hr prior to initiating the reaction. A mix containing RNA, ATP or AMPPNP, and RNAse inhibitor (all at 1.1x final concentration) was incubated at 20°C for 5 min prior to initiation with 1/10 vol of 100 nM enzyme. Reactions were quenched after the indicated incubation times by adding 10 μL of reaction to 5 μL of stop mix (0.3 % w/v SDS, 30 mM EDTA pH 8.0, 3 U/mL proteinase K [New England Biolabs]) followed by proteinase K digestion at 37°C for 1 hr and flash freezing in liquid nitrogen for storage at −80°C. Prior to gel loading, 15 μL of 89 mM Tris-borate pH 8.3, 7 M Urea, 2 mM EDTA, 12% w/v Ficoll, 0.005 % w/v xylene cyanol was added. The Mpp6 mix-in experiments were carried out in a similar fashion, except the indicated Mpp6 construct was included in the initial enzyme mix at 1.5 μM (1.5-fold molar excess). These reactions were incubated at 20°C for 1 or 8 min for Rrp6 and Rrp6exo-, respectively. 7.5 μL was loaded onto 15% acrylamide TBE urea gels (Life Technologies) to detect decay products. Gels were imaged for fluorescein fluorescence using a Typhoon FLA9500 instrument (GE).

## RNA binding assays

Fluorescence polarization experiments were performed by pre-incubating 50 nM 5' fluorescein labeled RNA on ice with increasing concentrations (0–6000 nM) of exoribonuclease dead proteins or complexes. The binding buffer consisted of 50 mM KCl, 20 mM Tris (pH8.0), 10 mM DTT, 0.5 mM $MgCl_2$, and 0.1% NP-40. Fluorescence polarization measurements were carried out as described previously (*Wasmuth and Lima, 2012*). Using data from triplicate experiments, a model for receptor depletion, when applicable, was used to calculate apparent $K_d$ values with Prism, GraphPad Software (RRID:SCR_002798).

## Yeast growth assays

Yeast growth media, including rich (YPD; yeast extract, peptone, dextrose) and minimal (SD) media, were prepared as described previously and standard approaches were used for the genetic manipulation of *Saccharomyces cerevisiae* (*Guthrie and Fink, 1991*).

All yeast pRS vectors were generated using standard cloning approaches (*Mumberg et al., 1995*). The *RRP6* open reading frame (ORF) and all *rrp6* alleles were PCR amplified from *S. cerevisiae* genomic DNA with oligonucleotides containing *EcoRI* and *SalI* restriction sites at the 5' and 3' ends, respectively, and then cloned onto a pRS413 vector. The wild-type *RRP6* allele was also cloned with the same restriction sites on a 2-micron pRS426 plasmid. All *rrp6* PCR constructs contain a stop codon at the 3' end. A promoter sequence corresponding to the first 129 base pairs (bp) upstream of the endogenous *RRP6* ORF was synthesized as a *XbaI/EcoRI* geneBlock (IDT) and cloned onto all *RRP6* and *rrp6* allele-containing pRS vectors. The *MPP6* ORF and all *mpp6* truncation alleles were generated as *BamHI/SalI* PCR fragments from yeast genomic DNA and cloned onto a pRS415 vector. All *mpp6* alleles contain a stop codon at their 3' end. A 500 bp fragment corresponding to the promoter sequence immediately upstream of the endogenous *MPP6* ORF was PCR-amplified using *NotI* and *BamHI* restriction sites on the 5' and 3' oligonucleotides, respectively, and inserted onto all pRS415 *MPP6* and *mpp6* vectors.

The haploid *S. cerevisiae* strains containing individual *rrp6Δ::KanMX* and *mpp6Δ::KanMX* gene deletions were obtained from the *Saccharomyces* Gene Deletion Project collection (Open Biosystems, Huntsville, AL) and are in the BY4741/BY4742 background (S288C, *his3Δ1 leu2Δ0 ura3Δ0 met15Δ0*). To generate the *rrp6 mpp6Δ* strain covered with the high-copy pRS426 *RRP6* plasmid, the MATa *rrp6Δ::kanMX* strain containing the pRS426 *RRP6* plasmid was mated with the MATα *mpp6Δ::KanMX* strain. The resulting heterozygous diploid was sporulated to obtain the *rrp6 mpp6Δ* + pRS426 *RRP6* double mutant by standard tetrad dissection. Both *rrp6Δ* and *mpp6Δ* deletions were verified by PCR genotyping using isolated genomic DNA and primer oligos flanking each *KanMX* deletion cassette. Presence of the pRS426 *RRP6* plasmid was confirmed with primers annealing within the *RRP6* ORF. We thank Marcel Hohl and the Petrini lab at MSKCC for assistance in constructing the *rrp6Δ mpp6Δ* double mutant. To obtain viable *rrp6 mpp6* strains in the presence of various combinations of pRS413 *rrp6* and pRS415 *mpp6* vectors, the *rrp6Δ mpp6Δ* + pRS426 *RRP6* strain was transformed with these plasmids and plated on selective SD -Ura-His-Leu solid medium. Individual clones were then streaked onto SD-His-Leu+5 FOA medium to select for viable strains lacking the pRS426 *RRP6* cover plasmid but maintaining the pRS415 and pRS413 vectors. Growth was monitored at 25°C and 30°C for 8 days.

Yeast growth via serial dilution was performed based on the protocol described by *Watts et al. (2015)*. All yeast strains were grown in selective SD -His-Leu medium overnight at 30°C. The next day, cells were diluted to an $OD_{600}$ of 1 before being spotted on SD -His-Leu or YPAD media in a ten-fold dilution series using a multichannel pipette (4 µl per spot). Cells were incubated at 30°C for 2 days and then imaged.

## Acknowledgements

This research was supported in part by the National Institute of General Medical Sciences of the National Institutes of Health under award numbers F31GM097910 (EVW), R01GM079196 (CDL), R35GM118080 (CDL), and P30 CA008748 (NIH NCI-Cancer Center Support Grant). The content is solely the responsibility of the authors and does not represent the official views of the National Institutes of Health. CDL is an investigator of the Howard Hughes Medical Institute. Work here was based in part upon research conducted at GM/CA@APS that has been funded in whole or in part with Federal funds from the National Cancer Institute (ACB-12002) and the National Institute of General Medical Sciences (AGM-12006). Work here was also based in part upon research conducted at NE-CAT beamlines (P41 GM103403, NIH NIGMS, S10 RR029205, NIH-ORIP HEI grant). Beamline research used resources of the Advanced Photon Source, a U.S. Department of Energy (DOE) Office of Science User Facility operated for the DOE Office of Science by Argonne National Laboratory under Contract No. DE-AC02-06CH11357.

## Additional information

### Funding

| Funder | Grant reference number | Author |
| --- | --- | --- |
| National Institutes of Health | F31GM097910 | Elizabeth V Wasmuth |

| | | |
|---|---|---|
| Howard Hughes Medical Institute | | Christopher D Lima |
| National Institutes of Health | R35GM118080 | Christopher D Lima |
| National Institutes of Health | R01GM079196 | Christopher D Lima |

The funders had no role in study design, data collection and interpretation, or the decision to submit the work for publication.

### Author contributions

EVW, Conceptualization, Resources, Data curation, Formal analysis, Funding acquisition, Investigation, Methodology, Writing—original draft, Writing—review and editing; JCZ, DZ, Resources, Data curation, Validation, Investigation, Methodology, Writing—review and editing; MD, Resources, Methodology, Writing—review and editing; CDL, Conceptualization, Data curation, Formal analysis, Supervision, Funding acquisition, Validation, Investigation, Visualization, Methodology, Writing—original draft, Project administration, Writing—review and editing

### Author ORCIDs

Christopher D Lima, http://orcid.org/0000-0002-9163-6092

## Additional files

### Major datasets

The following dataset was generated:

| Author(s) | Year | Dataset title | Dataset URL | Database, license, and accessibility information |
|---|---|---|---|---|
| Lima CD, Elizabeth V Wasmuth | 2017 | Atomic coordinates and structure factors | http://www.rcsb.org/pdb/explore/explore.do?structureId=5VZJ | Publicly available at the RCSB Protein Data Bank (Accession no: 5VZJ) |

The following previously published datasets were used:

| Author(s) | Year | Dataset title | Dataset URL | Database, license, and accessibility information |
|---|---|---|---|---|
| Lima CD, Wasmuth EV | 2014 | Structure of an Rrp6-RNA exosome complex bound to poly (A) RNA | http://www.rcsb.org/pdb/explore/explore.do?structureId=4OO1 | Publicly available at the RCSB Protein Data Bank (accession no: 4OO1) |
| Lima CD, Zinder JC | 2016 | Structure of an eleven component nuclear RNA exosome complex bound to RNA | http://www.rcsb.org/pdb/explore/explore.do?structureId=5K36 | Publicly available at the RCSB Protein Data Bank (accession no: 5K36) |

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
