## [Decision Letter]

Thank you for submitting your article "A Mpp6-nuclear exosome structure reveals interactions that stimulate the RNA exosome and recruit the Mtr4 helicase" for consideration by *eLife*. Your article has been reviewed favorably by three peer reviewers, and the evaluation has been overseen by Timothy Nilsen as Reviewing Editor and John Kuriyan as the Senior Editor. The following individuals involved in review of your submission have agreed to reveal their identity: David Tollervey (Reviewer #2); Andrzej Dziembowski (Reviewer #3).

The reviewers have discussed the reviews with one another and the Reviewing Editor has drafted this decision to help you prepare a revised submission.

All three reviewers were quite positive about the work but several issues need to be dealt with via revision. Please address these points thoroughly as possible; note that no additional experiments are required.

Essential revisions:

1) Figure 1: How were the contributions of Mpp6 to Dis3 and Rrp6 activities distinguished in the Exo11Dis3/Rrp6 complex? This is a key aspect of the work and it is a potential concern that the conclusions appear to have been based on a single RNA (polyA) that is not a major in vivo substrate.

2) Figure 1: "addition of Mpp6 diverts the pattern of UV-induced RNA crosslinks from Dis3 to Rrp6". This is a plausible interpretation of the protein gel, but the data would be more convincing is quantitated. As presented, it adds little to the paper.

3) Figure 1: Does RNA binding by channel occluded exosomes differ significantly from Mpp6 alone? Are we really just seeing the RNA binding affinity of Mpp6 here?

4) Figure 1 and Figure 4: How does Exo10Dis3 + Rrp6 + Mpp6 in Figure 4 differ from Exo11Dis3/Rrp6 + Mpp6 in Figure 1? The composition appears to be the same but the activity appears different – and more like Exo11Dis3/6-exo + Mpp6 in Figure 1.

Figure 4 also appears to indicate that Rrp6 does not contribute to the activity of the Exo10Dis3 + Rrp6 + Mpp6 complex, in contrast to a major conclusion of the manuscript that Mpp6 largely acts through Rrp6. Some clarification is needed

5) Figure 5: How was it established that "Mpp6 and Rrp47 stimulate Rrp6 activity" in the Exo11Dis3/Rrp6 complex used? Is this simply inferred from data in Figure 1?

6) The final sections on genetic interactions is an understandable attempt to relate the in vitro data to cellular phenotypes. However, it is not easy to follow the significance of the observations reported. This is not helped by the somewhat subjective scoring system used. For example: "Strains containing mpp6 (81–120) phenocopied Δmpp6". This appears to be a significant conclusion when stated, but is actually based on very limited data and, in fact, the + scores do not appear to be identical. These analyses would be much more valuable were the structural data used to identify specific amino acid changes that are predicted to yield suppression or synergistic negative interactions. As it is, the section adds little to the manuscript.

7) The information is densely packed in the text and sometimes a little difficult to follow. The text might need another round of editing to improve clarity.

8) In the section entitled 'Mpp6 can stimulate Rrp6 activity in an Exo9-dependent manner' the following first sentence of the paragraph does not connect well with the content of the subsequent sentences and needs a linking sentence "Mpp6 contains two small regions of high sequence conservation as noted previously (Milligan et al., 2008), one near its N-terminus and one near the middle of the protein."

---

## [Author Response]

*Essential revisions:*

*1) Figure 1: How were the contributions of Mpp6 to Dis3 and Rrp6 activities distinguished in the Exo11Dis3/Rrp6 complex? This is a key aspect of the work and it is a potential concern that the conclusions appear to have been based on a single RNA (polyA) that is not a major* in vivo *substrate.*

We assume the question regarding initial rates reported for Dis3 and Rrp6 from the Exo11^Dis3/Rrp6^ complex pertains to the complexity of these assays where both products are observed and where the products coalesce at the end of the time course. The calculation is based on the initial rates obtained from the early points of the time course where products of Rrp6 (intermediates) and Dis3 (final 4–5 nt products) are distinct and clearly separable by gel electrophoresis. We have also included labels in Figure 1 to explicitly indicate Rrp6 and Dis3 products to help the reader distinguish these activities. The procedures to separately quantify Dis3 and Rrp6 exoribonuclease have been established in Wasmuth and Lima, 2012 and reported in Wasmuth and Lima, 2017. We apologize for the confusion as some y axes were mislabeled in the prior submission, leading to the misconception that loss of substrate was being measured instead of accumulation of products.

The second comment pertains to our claim that Rrp6 is preferentially stimulated in an Exo11^Dis3/Rrp6^ complex. The reviewers point to an issue that this claim is supported solely by observations using a single polyA RNA, and that this RNA is not an in vivo substrate. While we agree that our synthetic RNAs cannot and were not intended as surrogates for in vivo substrates, we would argue that a polyA substrate is relevant because many RNAs are polyadenylated by TRAMP in the nucleus prior to degradation. Furthermore, we do include data for three other substrates, an AU-rich RNA (Figure 1—figure supplement 1) that is presumed to be more flexible than polyA RNA, and the RNA substrates in Figure 6 as a single stranded RNA substrate and as a partial duplex RNA substrate with a 3’ overhang. While we tested the activities of Exo11^Dis3/Rrp6^ with the AU-rich RNA, the RNA was degraded too quickly to ascertain rates when both enzymes were active, so we assessed the contributions of Mpp6 to degradation of AU-rich RNA by Exo10^Dis3^ and Exo10^Rrp6^. In both cases, Mpp6 stimulates the activities of Dis3 and Rrp6 in the context of the exosome complex (Figure 1—figure supplement 1, panels F and G). In Figure 6, we compare the activities of Exo11^Dis3/Rrp6^, Exo12^Dis3/Rrp6/Mpp6^, Exo12^Dis3/Rrp6/Rrp47^, and Exo13^Dis3/Rrp6/Mpp6/Rrp47^ with a mixed 27mer RNA 5’-CCCCACCACCAUCACUUAAAAAAAAAA alone and with a 17mer complementary RNA to generate a duplex RNA with a 10 nucleotide 3’ overhang. Comparing data in the presence of AMPPNP where Mtr4 is not active in the first row of gels to the last row in panel A shows that Rrp6 activities are stimulated by Mpp6 at the expense of Dis3 products using the ssRNA substrate, similar to results shown in Figure 1 with Exo11^Dis3/Rrp6^. In panel C, comparison of the leftmost column (Exo11^Dis3/Mpp6^) to the 3^rd^ column (Exo10^Dis3^) shows that Mpp6 can stimulate Dis3 to some extent in the absence of Rrp6, consistent with results shown in Figure 1 with polyA RNA and Figure 1 for AU-rich RNA.

*2) Figure 1: "addition of Mpp6 diverts the pattern of UV-induced RNA crosslinks from Dis3 to Rrp6". This is a plausible interpretation of the protein gel, but the data would be more convincing is quantitated. As presented, it adds little to the paper.*

We agree that this panel adds little to the paper. Cross-linking intensities are also prone to artifacts; more crosslinking does not necessarily mean more occupancy. While we interpreted more cross-linking to Rrp6 as a measure of RNA accessibility to Rrp6, it could simply be that Mpp6 shifts the RNA path slightly to bring the RNA closer to an amino acid that can form a crosslink and may not actually reflect the occupancy of the complex. We removed this figure from the paper.

*3) Figure 1: Does RNA binding by channel occluded exosomes differ significantly from Mpp6 alone? Are we really just seeing the RNA binding affinity of Mpp6 here?*

We show that Mpp6 alone binds RNA with a K_d_ > 1 µM (blue bar) and that Exo10^Channel-occ/Rrp6exo-/Mpp6^ binds RNA with a K_d_ of ~200 nM (orange bar), just two-fold less well than Exo10^Rrp6exo-/Mpp6^ (~90 nM; red bar), so we do not think that we are just seeing the RNA binding affinity of Mpp6 in Exo10^Channel-occ/Rrp6exo-/Mpp6^. Furthermore, Mpp6 can partially rescue Rrp6 activities in exosomes that include S1/KH channel mutations (Figure 1—figure supplement 1), suggesting that RNA binding activities of Mpp6 cooperate with additional RNA binding activities of the exosome (most likely the Rrp6 lasso) to stimulate Rrp6 even in the presence of channel mutations.

*4) Figure 4: How does Exo10Dis3 + Rrp6 + Mpp6 in Figure 4 differ from Exo11Dis3/Rrp6 + Mpp6 in Figure 1? The composition appears to be the same but the activity appears different – and more like Exo11Dis3/6-exo + Mpp6 in Figure 1.*

Thank you for catching this, we apologize for this error. The sample used in Figure 4 was not Exo11^Dis3/Rrp6^ (as in Figure 1) but rather Exo10^Dis3^ + Rrp6CTD or Rrp6^exo-^ (similar to Figure 1). This has been corrected.

*Figure 4 also appears to indicate that Rrp6 does not contribute to the activity of the Exo10Dis3 + Rrp6 + Mpp6 complex, in contrast to a major conclusion of the manuscript that Mpp6 largely acts through Rrp6. Some clarification is needed*

See above, this was an error in the figure.

*5) Figure 5: How was it established that "Mpp6 and Rrp47 stimulate Rrp6 activity" in the Exo11Dis3/Rrp6 complex used? Is this simply inferred from data in Figure 1?*

No, the claim that Mpp6 and Rrp47 stimulate Rrp6 activity was not solely inferred from data in Figure 1, but from data presented in Figure 1, Figure 5 and 6. In Figure 1 we analyzed the contribution of Mpp6 to Rrp6 activity, in Figure 5 we analyzed the contribution of Mpp6, Rrp47 or Mpp6/Rrp47 to Rrp6 activity, in Figure 6 we show analogous results with other substrates, all of which support our observation that Rrp6 activity can be stimulated by these two co-factors. An important distinction from previous literature is that we assay Rrp6 activities in reconstituted exosomes using full-length proteins rather than using truncated proteins or Rrp6 in isolation.

*6) The final sections on genetic interactions is an understandable attempt to relate the* in vitro *data to cellular phenotypes. However, it is not easy to follow the significance of the observations reported. This is not helped by the somewhat subjective scoring system used. For example: "Strains containing mpp6 (81–120) phenocopied Δmpp6". This appears to be a significant conclusion when stated, but is actually based on very limited data and, in fact, the + scores do not appear to be identical. These analyses would be much more valuable were the structural data used to identify specific amino acid changes that are predicted to yield suppression or synergistic negative interactions. As it is, the section adds little to the manuscript.*

While we agree that the scoring system is subjective, the statement that the ‘+ scores do not appear to be identical’ is incorrect. The scores for the pRS415 empty vector (i.e. mpp6Δ) in the originally submitted manuscript were, from left to right, -, ++++, +++, ++, -. The scores for the mpp6 (81–120) allele were also -, ++++, +++, ++, -. We appreciate the reviewer comments regarding the limited utility of data this figure as it pertains to the structure, but we argue for its inclusion as it shows various dependencies in vivo for domains characterized in vitro that were not apparent in the structure, so they have added value. For instance, mpp6 (1–120) can support growth in strains expressing an Rrp6 isoform that lacks the ability to interact with Rrp47, but only when the Rrp6 C-terminal tail is included. It also shows that cells harboring mpp6 alleles that can bind Mtr4 (full-length or 1–120) grow better in strains expressing an Rrp6 isoform that lacks the ability to interact with Rrp47 (rrp6 128–733 allele). This supports a major conclusion in our manuscript, that Mpp6 and Rrp47 both contribute to Mtr4 interactions, further corroborating results presented in Figure 5 and Figure 6. While we will defer to the reviewers and editors if they feel strongly on this point, we also feel strongly that this data should be included in the revised manuscript, as previous studies to this point have only identified Rrp47 as a bridge between Mtr4 and the exosome in yeast. We revised this section extensively to focus discussion on the most important observations to clarify these points.

*7) The information is densely packed in the text and sometimes a little difficult to follow. The text might need another round of editing to improve clarity.*

We made additional efforts throughout the manuscript in an attempt to streamline the presentation.

*8) In the section entitled 'Mpp6 can stimulate Rrp6 activity in an Exo9-dependent manner' the following first sentence of the paragraph does not connect well with the content of the subsequent sentences and needs a linking sentence "Mpp6 contains two small regions of high sequence conservation as noted previously (Milligan et al., 2008), one near its N-terminus and one near the middle of the protein."*

We revised this section and others to improve continuity.